# Environmental enrichment and physical exercise prevent stress-induced social avoidance and blood-brain barrier alterations via Fgf2

Sam E. J. Paton [1], José L. Solano [1], Alice Cadoret[1], Adeline Collignon [1], Luisa Bandeira Binder [1], Béatrice Daigle[1], Laura Menegatti Bevilacqua[1], Émanuelle Richer[1], François Coulombe-Rozon[1], Laurence Dion-Albert[1], Katarzyna A. Dudek [1], Signature Consortium*, Manon Lebel [1] & Caroline Ménard [1] ✉

Chronic stress promotes blood-brain barrier (BBB) integrity loss leading to passage of inflammatory mediators in mood-regulating brain areas and establishment of depressive behaviors. Conversely, neurovascular adaptations favoring stress resilience and preventive strategies to promote them are undetermined. We report that environmental enrichment dampens stress-induced loss of endothelial tight junction Claudin-5 (Cldn5) along with anxiety- and depression-like behaviors in male mice via an increase in fibroblast growth factor 2 (Fgf2). Coping with voluntary physical exercise also protects the BBB from stress deleterious effects by increasing Fgf2. Fgf2 is mostly expressed by glial cells, and viral-mediated astrocyte-specific *Fgf2* upregulation prevents stress-induced social avoidance while downregulation increases stress susceptibility and blunts physical exercise benefits. Treatment of mouse and human endothelial cells with Fgf2 prior an immune challenge reduces BBB dysfunction, Cldn5 loss, and altered signaling supporting its protective role. Circulating FGF2 level is linked with depression severity and symptomatology in men and women reinforcing involvement of this growth factor in mood disorders.

Major depressive disorder (MDD) is a psychiatric condition affecting >300 million people worldwide, representing a growing burden on global health systems[1]. Women are twice as likely to be diagnosed, and MDD presents sex differences in symptomatology, treatment responses and brain transcriptional profiles[2–5]. Common antidepressants are ineffective for 30–50% of depressed individuals suggesting that underlying causal mechanisms remain unaddressed[6,7]. First evidence of blood-brain barrier (BBB) leakiness in human depression by detection of brain-related proteins in the blood was reported 60 years ago[8]. Indeed, cerebrospinal fluid to

serum ratio of various peripheral markers are altered in individuals with MDD suggesting compromised BBB function[6,9,10]. The BBB is a dynamic frontier mediating communication between the periphery and the brain, composed of an intricate cellular network, including astrocytes, pericytes, and endothelial cells connected by specialized tight junctions[7,11]. This distinctive composition enables metabolic supply while also ensuring a selective permeability which protects the brain from bloodstream harmful toxins and inflammatory factors[12,13]. The BBB is necessary to maintain neural activity, and disruption can lead to neuroinflammation, neuronal death, and

[1]Department of Psychiatry and Neuroscience, Faculty of Medicine and CERVO Brain Research Center, Université Laval, Quebec, QC, Canada. *A list of authors and their affiliations appears at the end of the paper. ✉e-mail: Caroline.Menard@fmed.ulaval.ca

severe cognitive deficits[7]. For decades, mechanisms leading to neurovascular pathology in MDD remained elusive.

In the last years, several groups reported that in mice, chronic stress exposure, MDD main environmental risk factor, is associated with a sustained elevation of circulating inflammatory mediators such as interleukin-1β, interleukin-6 (IL-6) and tumor necrosis factor α (TNFα)[6,14]. This increase results in activation of brain endothelial cells via elevated cytokine expression, leukocyte adhesion, and degradation of tight junction protein Claudin-5 (Cldn5) leading to increased barrier permeability as assessed by passage of tracers[15–17]. BBB alterations and tight junction loss occur in a sex- and brain region-specific manner with the BBB being more vulnerable in the female prefrontal cortex (PFC) vs nucleus accumbens (NAc) for males[15,18]. The NAc is a forebrain nucleus playing key roles in reward and mood regulation while the PFC is involved in social behaviors, executive function and decision making[19,20]. Importantly, region-specific reduced expression of endothelial cell tight junction CLDN5 was confirmed in postmortem human brain samples from MDD men[21] and women[18] supporting a direct link between neurovascular health, stress vulnerability and human depression[7,15,16,18,22]. In other mental conditions like bipolar disorder and schizophrenia, the degree of BBB damage appears to correlate with age of onset as well as disease severity[23,24], suggesting that the neurovasculature could represent an innovative target for effective intervention against mood disorder development and progression.

Environmental conditions that promote stress resilience and neurovascular health offer a promising approach to identify therapeutic sites. Indeed, stimulating environments are well known to positively alter the adult brain, including vascularization and BBB function[25–28]. Further, circumstantial factors such as socioeconomic status and physical activity are negatively correlated with depression risk in humans[29,30], while in mice, access to nesting material, shelter, and toys (enriched environment, EE) or a running wheel (physical exercise, PE) attenuates depression-like behaviors following stress exposure[31,32]. The involvement of the BBB in these outcomes has not yet been assessed; however, we recently observed neurovascular changes after learning and memory tasks depending on environmental conditions[33]. The present study expands this idea by combining behavioral experiments performed in male and female mice with in vivo functional manipulations and in vitro cell signaling work to identify functional and transcriptional adaptations of the BBB involved in the pro-resilient effects of EE and PE during chronic stress. We report that environmental intervention can rescue stress-induced deficits in social behavior and expression of tight junction protein Cldn5 in both sexes. We identify fibroblast growth factor 2 (Fgf2) as a sex-specific protective factor upregulated in response to stress in the NAc of males with access to either EE or voluntary PE. Fgf2 is mostly expressed by astrocytes in both the mouse and human brain[34,35] and it is involved in neuronal and glial cell signaling, proliferation, and wound healing[36,37]. This growth factor can induce angiogenesis by stimulating endothelial cell proliferation in pathological context[38,39] and thus we hypothesized that it could be actively involved in pro-resilient neurovascular adaptations. Indeed, viral-mediated increase of astrocytic Fgf2 expression in the male NAc prevents stress-induced social avoidance, while downregulation in the same brain area promotes stress susceptibility. Fgf2 can dampen inflammatory activation of brain endothelial cells, as well as subsequent loss of barrier integrity by increasing Cldn5 expression, suggesting a potential protective mechanism. Finally, we associate a change in circulating FGF2 with depressive symptoms in human cohorts and highlight sex differences as well as the impact of university education, an important indicator of socioeconomic status.

## Results

### Environmental enrichment dampens stress-induced social avoidance and blood-brain barrier alterations in male mice

Social stress has been extensively studied in humans since higher prevalence of mood disorders and suicide attempts have been reported in victims of bullying[40]. The chronic social defeat stress (CSDS) paradigm is a commonly used mouse model of depression in which C57Bl/6 mice are exposed daily (5 min/day) for 10 days to a physical bout with a larger aggressive CD-1 mouse[41]. The day after the last stress bout, a social interaction (SI) test is performed, and behavioral phenotype is defined as the time spent in the interaction zone test when the aggressor is present divided by absent (Fig. 1A)[41]. In standard conditions with no supplements such as toys, shelter, or nesting material, about two-thirds of stressed mice display social avoidance (SI ratio <1) and are classified as stress-susceptible (SS), while the other third, showing no behavioral deficits, are considered resilient (RES) with a SI ratio >1[41]. To evaluate if EE has an impact on stress-induced BBB alterations, male C57Bl/6 were given access to nesting material, a shelter, and plastic chew toy on their side of the CSDS cage throughout the stress paradigm (Fig. 1A). EE increased the proportion of mice classified as resilient to more than half (Fig. 1B, stress x environment effect: **$p = 0.0029$, 57.9% with a SI ratio >1 vs 41% for standard housing conditions). As expected, almost all unstressed control (CTRL) mice were resilient (Fig. 1B). Importantly, stressed EE mice spent similar time in the corners during the SI test compared to unstressed controls, while mice subjected to CSDS in a standard environment developed social avoidance (Fig. 1C, stress × environment effect: ****$p < 0.0001$). No difference in exploration time was noted as measured by the distance traveled in the arena when the aggressor was either present or absent excluding locomotor issues (Supp. Fig. 1A).

To probe a role for the BBB in these positive effects, we next investigated stress-induced changes for transcription of BBB-related genes involved in cell proliferation, vascular remodeling, tight junction formation, or markers of astrocyte, pericyte, and neuroinflammation, in the NAc and PFC, two brain regions involved in the onset of depressive-like behaviors in mice and MDD in humans (Fig. 1D and Supp. Fig. 1B for behavioral data). Genes were selected based on previous work from our group and others showing changes in expression following stress exposure[15,18,42,43]. Standard CSDS reduces the expression of Cldn5, a key tight junction protein, specifically in the male NAc, and increases expression of inflammatory cytokine Il-6[15]. In contrast, we found in the NAc of stressed EE mice a general increase in gene expression associated with vascular remodeling and tight junction formation relative to unstressed EE control, including for Cldn5 (Fig. 1D, left, *$p = 0.0253$), as well as growth factors, particularly Fgf2 (*$p = 0.038$) which is linked to BBB integrity[44,45] and antidepressant behavioral effects[43,46–49]. No change in Cldn5 was observed in the male PFC in line with intact BBB integrity[15], but expression of several growth factors, including Fgf2 was upregulated (Fig. 1D, right, *$p = 0.0196$), suggesting adaptive mechanisms. The FGF family consists of ~20 ligands with a subset expressed in the brain. Most are involved in cortical patterning during development. Fgf2 is notable for its high expression in non-neuronal cells in mice and humans[34] (Supp. Fig. 1C), angiogenic properties[36,44], and implication in MDD[43,48–52]. We assessed the expression of Fgf1, 2, 4, 8, 9, 10, and 21, the most abundant of the FGF family, in the PFC and NAc (Supp. Fig. 1D). Fgf4, 8 and 21 were undetectable in both brain areas while Fgf9 and 10 expression levels could not be detected in cultured BBB-related astrocytes or endothelial cells (Supp. Fig. 1E) as expected from a cell-specific brain RNA-seq atlas[34]. Stress exposure had no impact on the expression of Fgf1 in the NAc (Supp. Fig. 1F). Comparison of NAc gene expression from standard CSDS and EE cohorts indicates a beneficial effect of EE through prevention of both Cldn5 loss (Fig. 1E, ****$p < 0.0001$) and increased inflammatory cytokine IL-6 (*$p = 0.0318$) following CSDS. Previous reports from standard CSDS show that SS and RES mice display substantially different transcriptional patterns[21]. Principal component analysis (PCA), used to identify strong patterns from the gene transcripts in our EE cohort, revealed that SS and RES mice strongly overlap (Fig. 1F). Indeed, when comparing gene expression patterns from our

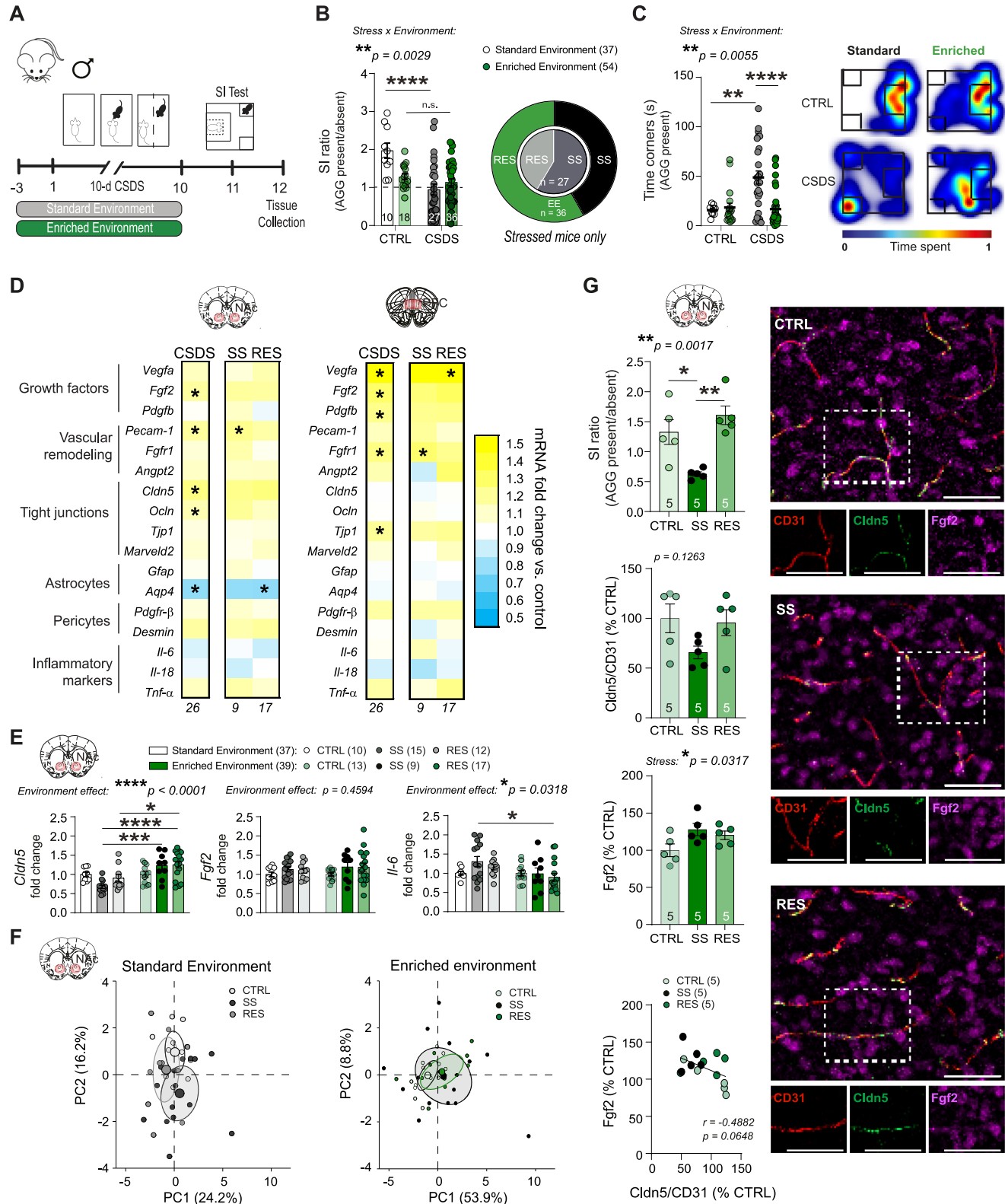

EE mice with previously published standard environment CSDS[15], we found evidence for close clustering of all behavioral phenotypes (CTRL, SS, and RES) based on BBB-related gene expression, in EE but not standard housed CSDS cohorts (Fig. 1F and Supp. Fig. 2D), indicating EE dampens stress effect on BBB-related transcription.

To confirm gene expression changes at the protein level, immunofluorescent staining was performed to label for Cldn5, Fgf2, and CD31, a marker of blood vessels, in brain slices from the NAc and PFC of

unstressed CTRL, SS and RES mice (Fig. 1G and Supp. Fig. 2A for behavioral data). In standard CSDS, Cldn5 coverage of blood vessels is reduced by ~50% in the NAc of SS male mice[15]. With EE, this loss is dampened and not significant anymore ($p = 0.1322$), supporting compensatory changes (Fig. 1G, Supp. Fig. 2B for CD31). In this line, while chronic stress has been reported to reduce *Fgf2* expression in a standard environment in the PFC[50], stressed mice in our EE cohort had increased Fgf2 in the NAc (*$p = 0.0317$), and this tends to correlate with

**Fig. 1 | Environmental enrichment dampens stress-induced social avoidance and blood–brain barrier alterations in male mice. A** Experimental timeline for chronic social defeat stress (CSDS) with enriched environment (EE). Male mice were housed with a nestlet, plastic chew toy, and shelter beginning 3 d prior to CSDS and continuing until the last defeat, followed by social interaction (SI) testing. **B** Compared to previously published results from CSDS with standard cages[15], stressed EE mice show less deficits in social behavior measured by the SI test, and a greater percentage of resilience. SI ratio was calculated by dividing the time spent in the interaction zone in presence vs absence of a novel CD-1 aggressor (AGG). Mice with SI < 1 were classified as stress-susceptible (SS), while SI > 1 were resilient (RES). **C** Stressed EE mice also show less time in corners of the SI test than those stressed in plain cages with no access to a nestlet, plastic chew toy, and shelter. Representative heatmaps of SI test in the second trial (AGG present) show differences between CTRL and CSDS mice with standard caging[15] and EE. **D** Heatmap showing transcription of BBB-related genes in the nucleus accumbens (NAc) and prefrontal cortex (PFC) after stress for EE mice. *Cldn5*, *Ocln*, and *Fgf2* are upregulated in the NAc of all stressed mice. **E** Increased *Cldn5* expression and decreased *Il-6* in SS EE mice compared to published data from SS mice in plain cages[15]. **F** CTRL, SS, and RES behavioral groups form distinct clusters based on principal component analysis of NAc gene expression data in standard CSDS but are grouped together in EE. **G** Fgf2 immunofluorescent labeling is increased in all stressed mice from the EE cohort, while Cldn5 relative to blood vessel area is diminished specifically in SS mice (scalebar = 50 μm). Fgf2 area correlates with the degree of Cldn5 loss suggesting a protective response. Data represent mean ± s.e.m., the number of animals is indicated on graphs. Group comparisons were evaluated with one- or two-way ANOVA followed by Bonferroni's post hoc tests and correlation with Pearson's correlation coefficient where appropriate; *$p < 0.05$, **$p < 0.01$, ***$p < 0.001$, ****$p < 0.0001$.

the degree of Cldn5 loss ($r = -0.4882$, $p = 0.0648$) (Fig. 1G). No change was observed in the PFC in EE conditions (Supp. Fig. 2C). Altogether, our findings suggest that access to an enriched environment has a protective effect on stress-induced BBB alterations, possibly via an elevation in Fgf2.

## Environmental enrichment rescues stress-related transcriptomic deficits in female mice

While CSDS is a standard protocol to induce social stress in male mice, it is not as relevant for female mice who do not commonly experience aggression in the wild[53]. Artificial methods for inducing social defeat in females exist, including application of male urine to promote aggressive bouts[54], but it introduces sensory information which may interfere with enrichment effects. In our hands, still only ~30% of female mice become susceptible to 10-day CSDS with this protocol, hampering the possibility to test preventive or protective approaches[18]. Thus, to evaluate whether EE could promote resilience in females, we took advantage of the subchronic variable stress (SCVS) paradigm, an established protocol producing anxiety and anhedonia after 6 days in females[18,55] along with BBB changes in the PFC[18]. Female mice assigned to control or SCVS groups had access to EE as described above and the SCVS group was exposed to a series of three repeated stressors, namely foot shock, tail suspension, and tube restraint (Fig. 2A). Following SCVS, a cohort was subjected to a battery of behavioral tests to assess the impact of EE on stress responses (Supp. Fig. 3A). As expected, exposure to SCVS in standard conditions induced anhedonia, a core feature of MDD (Supp. Fig. 3B, **$p = 0.0042$). Compared to 6-d SCVS with standard housing[18], access to EE prevented stress-induced reduction in time spent in the elevated plus maze (EPM) open arms (Fig. 2B, ***$p = 0.0002$), increased social interactions (*$p = 0.0297$), and sucrose preference (****$p < 0.0001$) (Supp. Fig. 3B-F). Exposure to 6-d SCVS is generally not sufficient to induce anxiety- or depression-like behaviors in males[56]. Nevertheless, to be consistent a cohort of male mice was subjected to the same stress paradigm without or with access to EE (Supp. Fig. 4A). No difference in behaviors was observed except for a significant effect of the environment for the SI ratio, regardless of stress exposure, due to increased time spent in the interaction zone when the CD-1 was absent (Supp. Fig. 4B-E). For tissue collection, a 2nd cohort of female EE + SCVS was tested in the EPM only to confirm normal behaviors despite stress exposure, then brain tissue was collected 24 h later (Fig. 2A and Supp. Fig. 5A, B for behavioral data) so 48 h after the last stressor, like for males (Fig. 1A). No difference in estrous cycle stage was observed between CTRL and SCVS groups at tissue collection (Supp. Fig. 5C).

Neurovascular disruption in the PFC underlies the development of anxiety- and depression-like behaviors in female mice and a loss of CLDN5 was noted in this brain area in postmortem samples from women with MDD[18]. Gene expression analysis showed that EE stabilizes BBB transcriptomic patterns in the PFC (Fig. 2C), leading to maintenance of *Cldn5* expression vs SCVS in standard housing (Fig. 2D,

**$p = 0.0024$) and normal behaviors despite stress exposure. We also observed that expression of *Fgf1* and *Fgf2* is similar in the NAc and PFC of female and male mice at baseline (Supp. Fig. 5D). Next, BBB-related gene expression patterns in the female PFC were compared after SCVS in standard vs EE conditions with PCA analysis. We revealed that with standard housing, CTRL and SCVS mice form distinct clusters, while with EE they are more closely grouped (Fig. 2E). Because EE reduces transcriptional differences between control and stressed mice in both males and females, gene expression patterns were compared in the NAc and PFC across sexes. Even if the behavioral outcomes are similar, BBB-related transcriptomic profiles of males and females differ, suggesting that environmental influence on the neurovasculature could be sex-specific (Fig. 2F, Supp. Fig. 5E, F). Given the implication of Fgf2 as a protective factor in males, we assessed its protein level along with Cldn5 and Cd31 in females. Immunofluorescent staining confirmed an absence of stress-induced changes in Cldn5 but also Fgf2 in both PFC (Fig. 2G, Supp. Fig. 5G) and NAc (Supp. Fig. 5H) in the female EE cohort. These results suggest that females benefit from an EE however, elevation of Fgf2 might be a male specific protective mechanism for pro-resilient effects on the BBB.

## Voluntary physical exercise protects the blood-brain barrier from the deleterious effects of stress, with light-cycle running promoting resilience

Fgf2 is sensitive to environmental conditions, and modifiable lifestyle factors such as physical exercise (PE) have been linked to elevated expression of this growth factor in the brain[57]. Voluntary PE has been proposed as a critical variable for disease prevention and stress resilience associated with environmental enrichment experiments[58]. PE has benefits for neurovascular health, however, it is unknown if it could protect from chronic social stress-induced BBB alterations. After running wheel habituation, male mice were then randomly assigned to either control or stress groups which both had free voluntary access to running wheels in their home cage throughout the CSDS protocol (Fig. 3A). Access to voluntary physical exercise during stress exposure increased the proportion of RES mice after CSDS to 53.6% (Fig. 3B), and like EE (Fig. 1C), strongly reduced social avoidance as measured by the time spent in the corners during the social interaction test (Fig. 3C, ****$p < 0.0001$ and Supp. Fig.6A-C for additional behavioral data). As expected, mice ran mostly during the dark cycle with distance increasing throughout the 10-d CSDS paradigm for the stressed mice to reach a total distance significant effect when compared to unstressed controls (Fig. 3D-E, *$p = 0.0241$). Intriguingly, RES mice ran more than other groups during the light cycle (Fig. 3F, **$p = 0.0061$), including right after the defeat bout (Fig. 3G, ****$p < 0.0001$ vs CTRL and *$p = 0.0136$ vs SS), suggesting that PE may represent an active coping strategy when facing social stress.

Brains were collected 48 h after the last stressor (Fig. 3A) and BBB-related genes in the NAc and PFC analyzed and compared between

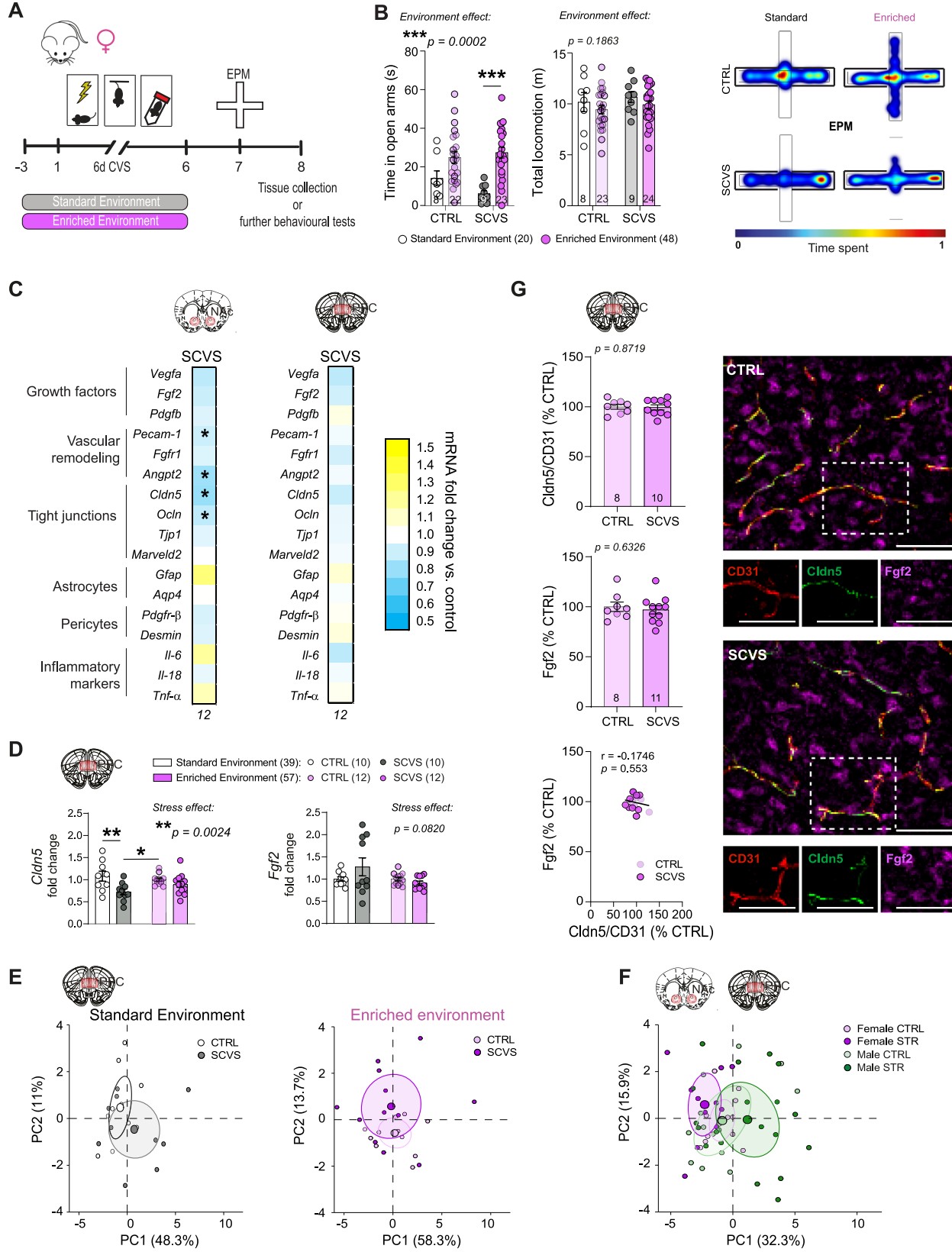

groups. Few changes were noted between CSDS and control mice with access to PE, with the exception of a strong increase in *Cldn5* transcript levels in both the NAc (**$p = 0.005$) and PFC (**$p = 0.0012$) of stressed mice and this effect was driven by RES animals (Fig. 3H, *$p = 0.0206$ for NAc and *$p = 0.0233$ for PFC). Conversely to our EE cohort where stressed mice displayed gene expression patterns distinct from

controls, PCA analysis of the PE CSDS cohort revealed that CTRL, SS, and RES mice all form similar clusters, though stressed mice exhibit more variance (Fig. 3I, Supp. Fig. 6E). At the protein level, CSDS did not affect Cldn5 levels in the NAc of our PE cohort, whereas all stressed PE mice exhibited an increase in Fgf2 staining compared to controls (Fig. 3J, Supp. Fig. 6D-G). Altogether, our results indicate a protective effect

**Fig. 2 | Environmental enrichment dampens stress-induced anxiety and blood-brain barrier alterations in the prefrontal cortex of female mice. A** Experimental timeline for subchronic variable stress (SCVS) with enriched environment (EE). Female mice were housed with a nestlet, plastic chew toy, and shelter beginning 3 days prior to stress and continuing until the last session, followed by elevated plus maze (EPM) and tissue collection 24 h later or further behavioral testing depending on the cohorts. **B** Compared to previously published results from female SCVS with plain cages and no access to a nestlet, plastic chew toy, and shelter[18], stressed EE mice show greater exploratory behavior characterized by open arm time in the EPM. Representative heatmaps show differences in EPM behavior between standard and EE SCVS. **C** Heatmaps showing transcription of BBB-related genes in the nucleus accumbens (NAc) and prefrontal cortex (PFC) after stress. *Cldn5* deficits are seen in the NAc, but not the PFC. **D** Increased *Cldn5* expression in stressed EE mice compared to published data from stressed mice in plain cages[18]. **E** CTRL and SCVS mice form distinct clusters based on principal component analysis of PFC gene expression data when performed in standard cages, but in the EE cohort they are more closely grouped. **F** BBB-related genes in the male NAc and female PFC respond differently to stress. **G** No changes in immunofluorescent staining of Fgf2 or Cldn5 following SCVS in female mice with EE (scalebar = 50 μm). Data represent mean ± s.e.m., the number of animals is indicated on graphs. Group comparisons were evaluated with two-way ANOVA followed by Bonferroni's post hoc tests or two-tailed t-tests with Welch's correction and correlation with Pearson's correlation coefficient where appropriate; *$p < 0.05$, **$p < 0.01$, ***$p < 0.001$.

---

of voluntary PE on stress-induced BBB changes in the male NAc, dampening social avoidance and favoring resilience.

## Upregulation of astrocytic *Fgf2* prevents stress-induced social avoidance

Since Fgf2 is mostly expressed by astrocytes in the mouse and human brain[34,35], and its level is increased in this cell type particularly along blood vessels in the context of physical exercise (Fig. 4A, B), we designed an adeno-associated virus (AAV) vector driving astrocytic *Fgf2* expression via the gfaABC1D promoter, and then tested its impact on social behaviors after CSDS exposure. First, we validated increased Fgf2 protein level in the NAc of AAV5-gfaABC1D-m*Fgf2*-P2A-GFP-injected mice when compared to animals injected with the control virus (AAV5-gfaABC1D-GFP) (Fig. 4C, D, left, **$p = 0.0067$). Intriguingly, the elevation in Fgf2 expression was mostly observed along blood vessels stained with the endothelial marker CD31 (Fig. 4D, right, *$p = 0.0163$). Next, a cohort of male mice was bilaterally injected with either the control AAV5-gfaABC1D-GFP or AAV5-gfaABC1D-m*Fgf2*-P2A-GFP viruses and then subjected to 10-day CSDS 4 weeks later to allow maximal *Fgf2* expression (Fig. 4E). Astrocytic increase in *Fgf2* did not alter the SI ratio, phenotype distribution (Fig. 4F, G) or locomotion (Fig. 4H), however, it reduced social avoidance as defined by the time spent in the corners of the arena (Fig.4I). Stressed mice spent more time in the corners, an effect driven by the AAV5-gfaABC1D-GFP-injected animals (Fig.4I, left, *$p = 0.0407$) with no difference observed for AAV5-gfaABC1D-m*Fgf2*-P2A-GFP mice ($p = 0.5267$). In fact, only SS mice injected with the control virus displayed social avoidance when compared to AAV5-gfaABC1D-GFP unstressed control and RES groups (Fig. 4I, right, **$p = 0.0089$ for SS vs CTRL; *$p = 0.0453$ for RES vs SS) or AAV5-gfaABC1D-m*Fgf2*-P2A-GFP-injected animals categorized as SS due to a SI ratio <1 (*$p = 0.0258$). Tissue collection after behavioral assessment confirmed higher *Fgf2* level in the NAc of mice injected with the AAV5-gfaABC1D-m*Fgf2*-P2A-GFP virus (Fig. 4J). These results suggest that increasing *Fgf2* in astrocytes can prevent CSDS-induced social avoidance, mimicking the beneficial effect of access to an enriched environment (Fig. 1C) or voluntary physical exercise (Fig. 3C) while facing chronic social stress exposure.

## Downregulation of astrocytic *Fgf2* expression induces social stress susceptibility and blunts the benefits of physical exercise

Next, to confirm that elevation of astrocytic *Fgf2* is involved in stress resilience, we designed an AAV vector with a short harpin RNA (shRNA) blocking *Fgf2* expression under the gfaABC1D astrocyte-specific promoter (Fig. 5A). We first validated decreased Fgf2 protein level in the NAc of AAV5-gfaABC1D-shRNA-m*Fgf2*-GFP-injected mice when compared to animals injected with the control virus (AAV5-gfaABC1D-shRNA-scramble-GFP) in a cohort that did not go through behavioral testing (Fig. 5B, *$p = 0.0163$). A cohort of male mice was bilaterally injected with either the control AAV5-gfaABC1D-shRNA-scramble-GFP or AAV5-gfaABC1D-shRNA-m*Fgf2*-GFP viruses and then subjected to 10-day CSDS 4 weeks later to allow maximal viral expression (Fig.5C).

Reducing astrocytic *Fgf2* in the NAc altered the SI ratio after CSDS exposure, with a higher proportion of SS mice (Fig. 5D, E, *$p = 0.0499$). This was not due to a change in locomotion (Fig. 5F). Another cohort of mice was bilaterally injected with either the control AAV5-gfaABC1D-shRNA-scramble-GFP or AAV5-gfaABC1D-shRNA-m*Fgf2*-GFP viruses and subjected to 10-day CSDS 4 weeks later but with access to running wheels (Fig. 5G). Stressed mice from both groups displayed social avoidance, as measured by lower SI ratio (***$p = 0.0005$) and higher number of entries in the corners when the AGG was present (*$p = 0.0223$), supporting our hypothesis that an elevation in NAc astrocytic *Fgf2* may contribute to the benefits of PE (Fig. 5H, I). Behavioral alterations were not due to impaired locomotion (Fig. 5J). In fact, stressed mice injected with the control virus ran increasing distance across days after the first defeat bout (Fig. 5K) as observed in non-injected animals (Fig. 3D). This was not the case for AAV5-gfaABC1D-shRNA-m*Fgf2*-GFP-injected mice (Fig. 5L), suggesting that motivation could be affected, with the NAc playing a crucial role in reward processing[19]. In line with these behavioral findings, we observed an elevation of *Fgf2* expression in the stressed PE cohort injected with the control AAV5-gfaABC1D-shRNA-scramble-GFP virus (**$p = 0.0024$) (Fig. 5M). In contrast, AAV-driven reduced *Fgf2* expression was similar in the NAc of AAV5-gfaABC1D-shRNA-m*Fgf2*-GFP-injected mice without (**$p = 0.0025$) or with access to running wheels (***$p < 0.0001$) (Fig. 5M), indicating that Fgf2 may be involved in the rewarding effects of PE.

## In vitro treatment with Fgf2 reduces TNF-α-induced Cldn5 loss, endothelial cell signaling alterations, and barrier hyperpermeability

BBB disruption is associated with stress-induced behavioral deficits in mice[15,16,18,22] and psychiatric disorders, including MDD in humans[15,18,21,23,24,59]. Fgf2 can mediate the formation of tight junctions in endothelial cells[44,45], nevertheless, it is undetermined if Fgf2 could protect against stress-related neurovascular damage. Our findings indicate a role for Fgf2 in protective effects of EE on the NAc BBB, specifically in male mice, with upregulation of this growth factor occurring in parallel with increased Cldn5 expression at both gene and protein level in stressed mice with access to EE (Fig. 1). Therefore, we investigated whether Fgf2 can preserve brain endothelial cell properties in vitro using treatment with the proinflammatory cytokine TNF-α as a biological stressor. Indeed, TNF-α is elevated in the blood of mice after CSDS and individuals with MDD[60–62], while downstream signaling through TNF receptors is linked to stress-induced BBB breakdown in mice[21,22]. To ensure translational relevance we exposed both human (HBEC-5i) and mouse (bEnd.3) brain endothelial cell lines to either acute (<24 h) or chronic (up to 7 days) periods of inflammatory challenge with 10 ng/mL of TNF-α (Fig. 6A). Doses of TNF-α and timepoints for cell collection were determined according to previous publications describing impact of this cytokine treatment on endothelial cells[63–65]. Cells were pretreated for 1 h with FGF2 (10 ng/mL[38]) to mimic habituation with EE/PE, before co-stimulation with TNF-α and/or FGF2.

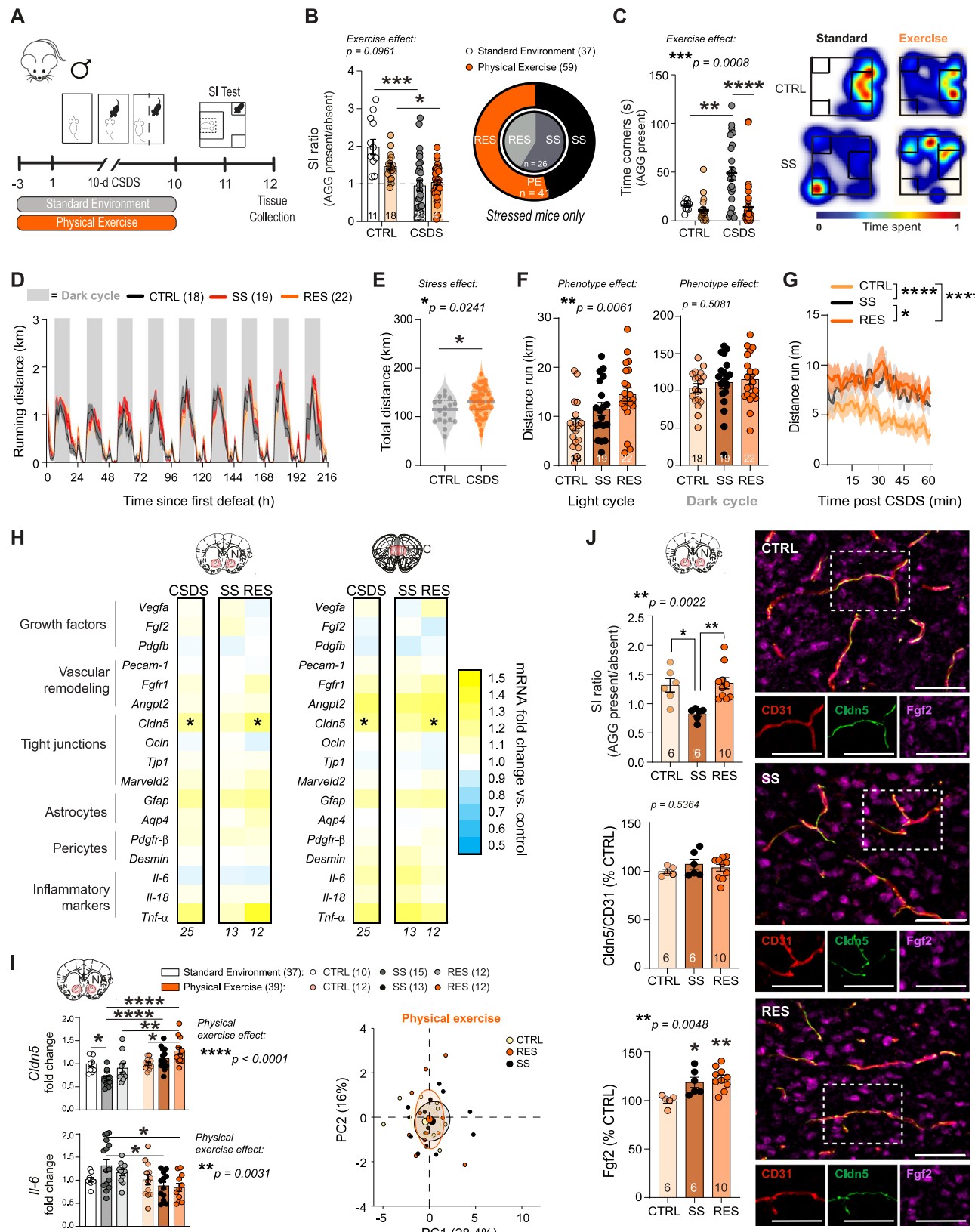

Following TNF-α treatment, expression of genes associated with tight junctions, FGF2 signaling, and proinflammatory activation of endothelial cells was evaluated at several timepoints (1, 3, 6, or 24 h). Acute TNF-α stimulation induced strong endothelial cell activation characterized by downregulation of tight junction proteins *Cldn5/CLDN5* and *Ocln/OCLN*, as well as expression of inflammatory factors such as

*IL-6* (HBEC-5i) and *Vcam-1* (bEnd.3) (Fig. 6B, C). In both cell types, FGF2 attenuated TNF-α-induced loss of *CLDN5*, with protective effects especially prominent after 6 h of treatment (Fig. 6D, E, *p = 0.0119 for mouse and ***p = 0.0004 for human endothelial cells).

Since FGF2 could reverse the effects of acute TNF-α treatment on *CLDN5* expression, we next tested if it could prevent loss in barrier

**Fig. 3 | Physical exercise protects the blood-brain barrier from deleterious effects of stress with light-cycle running promoting resilience. A** Experimental timeline for chronic social defeat stress (CSDS) with physical exercise (PE). Male mice were habituated with a battery-powered running wheel prior to CSDS and had voluntary access to wheel running until the last defeat, which was followed by social interaction (SI) testing. **B** Compared to previously published results from CSDS with plain cages[15], stressed PE mice show similar deficits in social behavior measured by the SI test, but a greater percentage of resilience. SI ratio was calculated by dividing the time spent in the interaction zone in presence vs absence of a novel CD-1 aggressor (AGG). Mice with SI < 1 were classified as stress-susceptible (SS), while SI > 1 were resilient (RES). **C** Stressed PE mice show substantially less time in corners of the SI test than those stressed in plain cages with no running wheel (n = 11–41). Representative heatmaps of SI test in the second trial (aggressor present) show differences between CTRL and SS mice in CSDS with standard caging[15]

and EE. **D** Representative graph showing running activity per hour throughout CSDS. **E** Stressed mice run slightly more than controls. **F** Stress phenotype is associated with running during the light cycle, with RES mice running more during the day. **G** RES mice run more in the hour following stress (****p < 0.0001).
**H** Heatmaps showing transcription of BBB-related genes in the nucleus accumbens (NAc) and prefrontal cortex (PFC) after stress with PE. *Cldn5* is upregulated following 10 d CSDS in both brain regions of mice with access to PE. **I** Increased *Cldn5* expression and decreased *Il-6* in SS EE mice compared to published data from SS mice in plain cages. **J** No loss of Cldn5 immunofluorescent labeling in SS mice with PE access (scalebar = 50 μm). Fgf2 immunofluorescent labeling is increased in all stressed mice from the PE cohort. Data represent mean ± s.e.m., the number of animals is indicated on graphs. Group comparisons were evaluated with one- or two-way ANOVA followed by Bonferroni's post hoc tests, or two-tailed t-tests with Welch's correction; *p < 0.05, **p < 0.01, ***p < 0.001, ****p < 0.0001.

integrity following long-term TNF-α exposure. Chronic stimulation with TNF-α over 7 days altered endothelial monolayer integrity as measured with trans-endothelial electrical resistance (TEER) when compared to control wells, by day 1 in bEnd.3 and day 3 in HBEC-5i, and this effect was prevented by FGF2 co-treatment (Fig. 6F, G, **p = 0.0014 for mouse and ****p < 0.0001 for human endothelial cells). FGF2 is a potent mitogen thus, an MTT assay was conducted to rule out potential changes in cell number which could influence TEER. Once confluent, FGF2 did not influence cell number vs controls (Supp. Fig. 7). On day 7 of the TEER protocol, cells were fixed and stained for Cldn5 to visualize tight junctions. bEnd.3 cells treated with TNF-α showed disruption of tight junction structure, including presence of spikes, discontinuities, and membrane ruffling which are signs of endothelial dysfunction[66]; however, this morphology was not observed for cells co-stimulated with Fgf2 (Fig. 6H).

## Fgf2 induces GSK3β phosphorylation and prevents β-catenin dissociation from tight junctions

To further gain mechanistic insights, the molecular mechanisms underlying the rescue of TNF-α-induced Cldn5 loss by Fgf2 were investigated. An integral part of TNF-α response in endothelial cells is activation of the Akt/ERK pathways, which slow and eventually terminate the inflammatory signaling cascade[65]. Fgf2 is a potent activator of these pathways[67], and we evaluated whether the protective effects of Fgf2 on endothelial cells could be a result of rapid resolution of inflammatory signaling. Akt phosphorylates GSK3β serine residues, inhibiting this redox-sensitive enzyme and crucial mediator of TNF-α signaling known to disrupt tight junction integrity when activated in endothelial cells[63,64,68]. Therefore, we performed western blotting to assess the effects of TNF-α and Fgf2 on phosphorylation of GSK3β in the human HBEC-5i cell line (Supp. Fig. 8). We found that 1 h pre-treatment with FGF2 induced GSK3β serine-9 phosphorylation (Fig. 7A, *p = 0.0332), but it did not prevent the rapid (5 min) dephosphorylation response upon TNF-α stimulation (Fig. 7A). β-catenin is a downstream target of GSK3β, which normally interacts with cadherins to promote tight junction integrity, but in the context of stress and inflammation, it can be internalized to promote deleterious signaling[21,69,70]. In concordance with GSK3β inhibition, 1 h of FGF2 treatment tended to reduce serine/threonine β-catenin phosphorylation (Fig. 7B, p = 0.0747). However, while TNF-α treatment initially reduced β-catenin phosphorylation (**p = 0.0031), when co-administered with Fgf2 it led to a strong increase in β-catenin phosphorylation peaking 1 h following TNF-α introduction (***p = 0.0004) (Fig. 7B). These results suggest that FGF2 can regulate β-catenin dynamics during inflammatory activation of brain endothelial cells which could mitigate stress-induced BBB alterations (Fig. 7C).

With endothelial β-catenin signaling essential for tight junction regulation and BBB integrity, we assessed if changes in inflammation-induced β-catenin phosphorylation mediated by FGF2 treatment could affect β-catenin cellular distribution and tight junction morphology. β-

catenin was stained by immunofluorescence in human HBEC-5i cells in control conditions or following 30 min of TNF-α treatment with or without FGF2. TNF-α-treated cells displayed tight junction spikes and discontinuities indicative of ultrastructural disruption as well as possible β-catenin internalization, whereas this morphology was not observed in control or TNF-α/FGF2 co-treated cells (Fig. 7D, E). Furthermore, TNF-α induced distension of β-catenin-labeled tight junction strands compared to vehicle-treated wells, as the width of β-catenin labeled tight junctions, was increased in TNF-α-treated wells (Fig. 7D, E). This diffusion was normalized when cells were co-administered FGF2, suggesting a role for this growth factor in stabilizing β-catenin at sites of cell adhesion and preserving normal tight junction morphology. Since both Fgf2 and β-catenin have been implicated in vascular remodeling and wound repair[71,72], a scratch wound test was performed to evaluate endothelial healing responses after TNF-α treatment (Fig. 7F). As expected, TNF-α substantially reduced wound healing (Fig. 7F). Interestingly, while FGF2 alone did not affect wound repair compared to control, it did improve TNF-α-related reduction in healing rate to result in a higher percentage of total healed area after 12 h (Fig. 7F). This finding suggests that this growth factor not only attenuates TNF-α-induced inflammatory signaling in endothelial cells, but it can restore functional and healing properties of the BBB which may underlie beneficial impact of EE in preventing stress-induced neurovascular alterations and promoting resilience.

## The environment is a key factor determining BBB response to stress in mice and MDD pathogenesis in humans

To this day, mood disorders, including MDD, are still diagnosed with questionnaires only. Identification of biomarkers with potential to inform clinicians for diagnosis and treatment choice is greatly needed and blood immune and vascular markers have received increasing attention in recent years[6,18,73–75]. To validate the importance of environmental conditions in stress-induced behavioral responses and underlying biology, we first conducted a hierarchical clustering analysis of all our mouse cohorts for BBB gene expression in stress-sensitive brain areas. It revealed that the environment is a key factor influencing BBB gene expression in both male (Fig. 8A) and female (Fig. 8B) mice. Indeed, groups tend to cluster by environmental conditions as opposed to stress exposure or phenotype. With this in mind, we explored if circulating FGF2 could be associated with human MDD when considering socioeconomic factors. Clinical evidence and post-mortem studies related to FGF2 in the context of MDD are inconsistent[51,52,76–80] and this could be due to sampling heterogeneity. Blood serum FGF2 level was measured by ELISA with a high sensitivity kit in samples from men and women with a diagnosis of MDD and compared to matched controls. MDD diagnosis and severity of symptoms had a significant impact on FGF2 level (Fig. 8C, D, *p = 0.026), which may explain previously reported discrepancies. Socioeconomic position, as measured by education, income, and

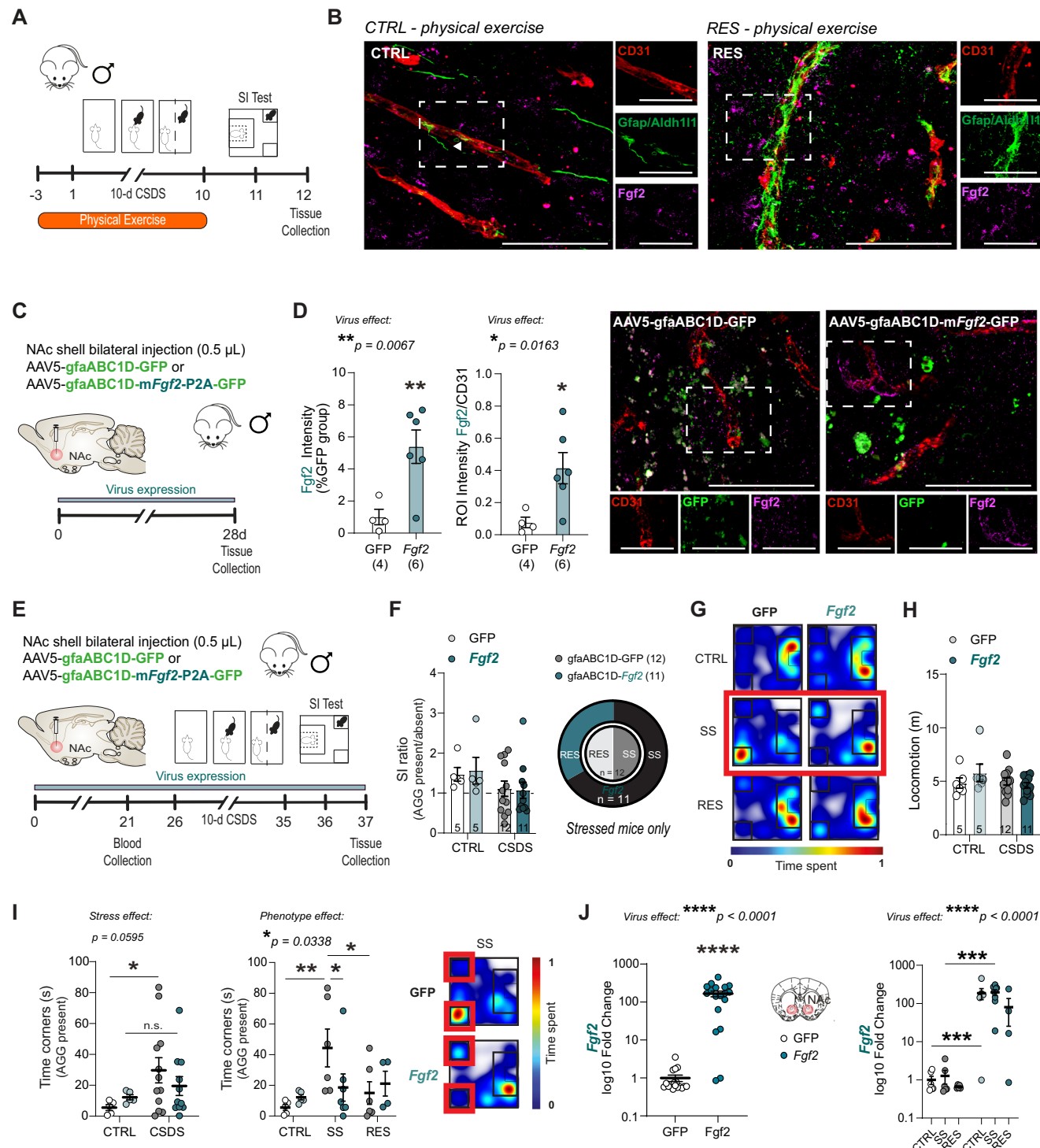

**Fig. 4 | Upregulation of astrocytic *Fgf2* prevents stress-induced social avoidance. A** Experimental timeline for chronic social defeat stress (CSDS) with physical exercise (PE). Male mice were habituated with a battery-powered running wheel prior to CSDS and had voluntary access to wheel running until the last defeat, which was followed by social interaction (SI) testing and tissue collection 24 h later. **B** Increased Fgf2 and astrocyte marker immunoreactivity is observed along blood vessels of resilient mice with access to PE when compared to unstressed PE controls (scalebar = 50 µm and 25 µm for high magnification images). **C** Experimental timeline for AAV5-gfaABC1D bilateral injection in the NAc. Mice were injected and 4 weeks later, NAc brain tissues were collected. **D** Mice injected with AAV5-m*Fgf2*-P2A virus have higher Fgf2 signal intensity (left), precisely colocalized on blood vessels (CD31), compared to controls AAV5-GFP (middle). Fgf2 signal is strongly colocalizing with CD31+ endothelial cells (representative images on the right) (scalebar = 50 µm and 25 µm for high magnification images). **E** Experimental

timeline for AAV5-gfaABC1D bilateral injection in the NAc, followed by CSDS, then Social Interaction (SI) test. **F** SI ratio is similar between AAV5-GFP and AAV5-m*Fgf2*-P2A, either for control unstressed mice (CTRL, *n* = 5/group) or stressed mice (CSDS, *n* = 12/group). AAV5-m*Fgf2*-PA2 mice from the stressed group show higher susceptibility compared to AAV5-GFP stressed mice (**G**) no difference in locomotion (**H**) however, they spend more in the interaction zone (**I**) and less time in the corners when the aggressor (AGG) is present, particularly in susceptible (SS) mice that received AAV5-m*Fgf2* virus (*n* = 5–12/group). **J** After CSDS, AAV5-m*Fgf2* mice show high levels of *Fgf2* mRNA, no matter their CSDS group, compared to mice with control virus AAV5-GFP. Data represent mean ± s.e.m., the number of animals is indicated on graphs. Group comparisons were evaluated with one- or two-way ANOVA followed by Bonferroni's post hoc tests, or two-tailed t-tests with Welch's correction; *p < 0.05, **p < 0.01, ***p < 0.001, ****p < 0.0001.

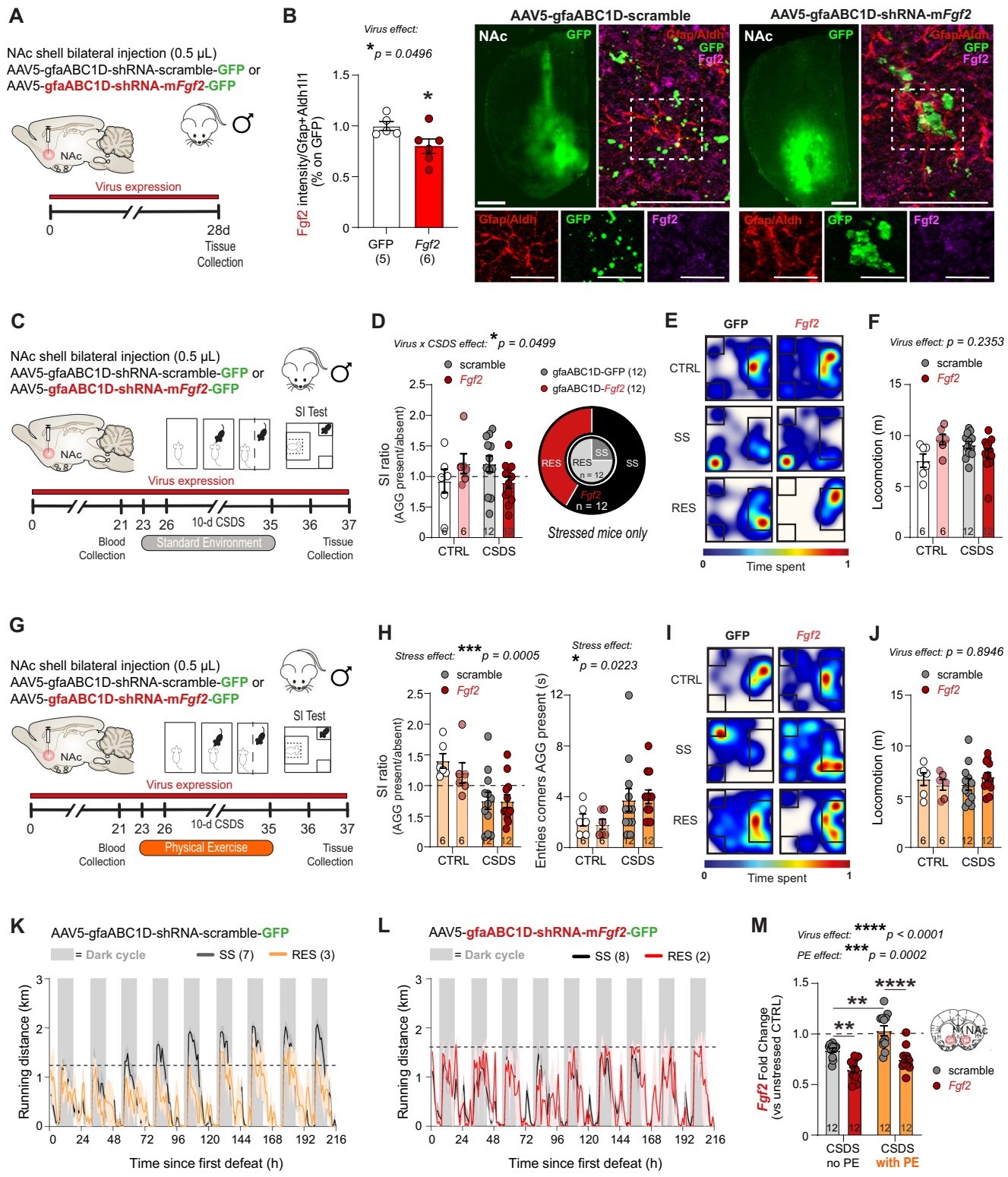

occupational prestige, is a strong determinant of MDD onset[81]. Accordingly, in our cohort, having a university diploma and being employed was associated with lower depression rates (Fig. 8E, F) so we tested the impact on FGF2 blood level. In individuals with no university diploma, an increase in circulating FGF2 was measured for MDD samples (Fig. 8E, **p = 0.006). An elevation in circulating FGF2 could be linked to MDD-associated neurovascular alterations and an attempt to repair the leaky BBB[82]. Importantly, a change in circulating FGF2 was associated with MDD in all men as assessed by the overall Patient

Health Questionnaire (PHQ-9) score, a widely used screen test for depression (Fig. 8G, right). Blood level of FGF2 was significantly correlated with multiple symptoms of this psychiatric condition in both women and men (Fig. 8G). Sex-specific symptomatology was noted with men with MDD reporting depressed mood, insomnia, and devaluation while women mentioned anhedonia, fatigue, a change in appetite, and concentration capacities, all correlated with changes in blood FGF2 (Fig. 8G). To sum up, circulating FGF2 may represent a promising blood biomarker of mood disorders, however, it is highly

**Fig. 5 | Downregulation of astrocytic *Fgf2* in the NAc increases stress susceptibility and blunts physical exercise benefits. A** Experimental timeline for AAV5-gfaABC1D-shRNA bilateral injection in the NAc. Mice were injected and 4 weeks later, NAc brain tissues were collected. **B** Mice injected with AAV5-shRNA-m*Fgf2* virus have lower Fgf2 signal intensity in astrocytes compared to controls AAV5-GFP (representative images of NAc infected area on the left, scalebar = 2 mm, and cell specificity on the right, scalebar = 50 μm with 25 μm for high magnification images below). **C** Experimental timeline for AAV5-gfaABC1D-shRNA bilateral injection in the NAc, followed by chronic social defeat stress (CSDS), then Social Interaction (SI) test. **D** SI ratio is lower for stressed mice with downregulation of astrocytic *Fgf2* in the NAc. AAV5-gfaABC1D-shRNA-m*Fgf2* mice from the stressed group display higher susceptibility compared to AAV5-gfaABC1D-shRNA-scramble stressed mice (**E**) but no difference in locomotion (**F**). **G** Experimental timeline for AAV5-gfaABC1D-shRNA bilateral injection in the NAc, followed by CSDS with access to running wheels, then SI test prior tissue collection. **H** Chronic social stress promotes social avoidance as measured by lower SI ratio and increased number of entries in the corners in both AAV5-gfaABC1D-shRNA-mFgf2- and AAV5-gfaABC1D-shRNA-scramble-injected mice (**I**) and this is not related to locomotion impairment (**J**). **K** Running distance increases across days for mice injected with the control virus but not for AAV5-gfaABC1D-shRNA-m*Fgf2*-injected animals (**L**). The dotted line is set at maximum running distance during the first 24 h. **M** After CSDS, AAV5-gfaABC1D-shRNA-m*Fgf2* mice show lower levels of *Fgf2* mRNA without or with access to physical exercise (PE), compared to mice injected with control AAV5-gfaABC1D-shRNA-scramble virus. PE also elevates *Fgf2* in this group. Data represent mean ± s.e.m., the number of animals is indicated on graphs. Group comparisons were evaluated with one- or two-way ANOVA followed by Bonferroni's post hoc tests, or two-tailed t-tests with Welch's correction; *$p < 0.05$, **$p < 0.01$, ***$p < 0.001$, ****$p < 0.0001$.

## Discussion

Identifying, creating, and sustaining stimulating environments is a crucial strategy to ease the immense burden of mood disorders worldwide[83]. Environmental factors like socioeconomic status and physical exercise are negatively associated with MDD risk in humans[29,30], while access to complex housing or voluntary exercise via running wheels have been highlighted as protective strategies against maladaptive stress responses in mice[31]. Enrichment elicits changes in synaptic plasticity, growth of new neurons, and epigenetic modifications in neuronal populations[84], however the mechanisms linking environmental features to brain biology are complex, and their impact on non-neuronal cells is not well understood. The BBB is a crucial interface for communication between the environment and the brain, and disruption of this barrier is implicated in pathogenesis of stress-related mood disorders, including depression[15,18]. Here, we demonstrate that housing conditions modify the neurovascular response to stress and influence social, anxiety- and depressive-like behavior in both male and female mice. Access to structural enrichment or physical exercise during CSDS attenuated stress-induced loss of *Cldn5* gene expression and tight junction coverage of blood vessels in the male NAc, a brain region involved in emotional regulation and mood disorder pathophysiology. Further, we report that protective effects of home cage enrichment on the neurovasculature in male, but not female mice coincide with elevated Fgf2, a ubiquitous growth factor known to have anxiolytic and antidepressant effects[43,46–49], in the NAc. Beneficial effect of Fgf2 on the brain vasculature when facing a chronic social stress challenge was confirmed with voluntary physical exercise, known to increase production of this growth factor. To see whether this increase could protect the BBB, we combined in vivo viral-mediated functional manipulations with in vitro models and showed that increasing *Fgf2* expression in astrocytes prevents CSDS-induced social avoidance and treatment with this growth factor attenuates downregulation of *Cldn5* expression and preserves endothelial monolayer integrity upon treatment with TNF-α in both mouse and human brain endothelial cells. Finally, we found an association for circulating FGF2 level with depression severity and symptomatology in human blood samples from men and women with a diagnosis of MDD with an impact of socioeconomic factors in line with our mouse findings.

Previous studies have reported protective effects of complex environment or physical activity on the BBB in a variety of disease models, including vascular dementia, multiple sclerosis, and Alzheimer's disease[85–88]. Here, we show that EE or PE can also protect BBB properties against a purely psychological stressor. In males, EE was associated with a broad reconfiguration of stress-induced transcriptional patterns at the BBB, including upregulation of *Fgf2* growth factor expression, which is associated with maintenance of blood vessel integrity and maintenance of Cldn5 expression[45,89], and appears to drive EE-mediated change in *Cldn5* in response to stress. In previous RNA-seq comparisons of endothelial cell transcription from male mice after CSDS, SS and RES mice shared very few overlapping changes in gene expression[21], while the subset of transcripts assessed from our EE cohort showed SS and RES mice had many differentially expressed genes in common. These changes imply that instead of simply preventing stress-related damage in males, EE actively reconfigures the BBB transcriptional response to CSDS, which was confirmed by comparing the clustering of CTRL, SS, and RES mice based on transcription of BBB-related genes in standard CSDS or with access to EE. Similarly, access to PE during chronic stress also resulted in upregulation of *Cldn5* mRNA expression along with Fgf2 protein staining in the male NAc. However, instead of broadly altering the transcriptional response to stress, PE more precisely targets upregulation of *Cldn5* mRNA expression in stressed mice. These findings correspond with several lines of evidence suggesting that exercise and enrichment exert distinct effects on the brain and body[84,90,91]. Furthermore, it has been argued that exercise is a crucial factor for the beneficial effects of environmental enrichment[58], and our results suggest this could be due to specific upregulation of *Cldn5* at the BBB to protect against damage. Various structural rodent EE have been described in the past decades[28,92]. The 10-d CSDS protocol, as employed here, is performed under conditions that can be considered impoverished (no nestlet, house or toys). This is necessary to prevent transfer of stained material, nesting for example, with unfamiliar urine odor known to exacerbate aggressive behaviors[93]. The recent development of complex semi-natural environments with tracking systems and machine-learning assisted analyses of video recordings[94,95] will allow to explore further the neurobiological mechanisms underlying naturalistic social behaviors, it will be intriguing to assess how it will impact BBB biology.

The use of different stress protocols in each sex means we could not directly compare the effectiveness of environmental interventions between males and females. This study aimed to investigate the effects of environment on stress resilience and thus requires a strong stress effect for which a rescue can be assessed. In female mice, CSDS is less applicable as females in the wild do not experience social aggression in the same way as males, and CD1 aggressors are moreover less likely to attack female intruders[53]. Several attempts have been made to overcome these barriers by chemogenetically inducing aggression in CD1s, or applying male urine to the back of female mice, in order to provoke an attack, but these methods still only succeed in producing stress-susceptible behaviors in about 1/3 of the stress cohort[18,54]. This suggests a ceiling for susceptibility and a potential barrier for detecting pro-resilient effects of enrichment. Thus, we opted to use instead a chronic variable stress protocol, which is well validated and produces strong behavioral alterations in female mice that correspond, to some extent, to human depression[55,56]. As described before[56], 6-day SCVS led

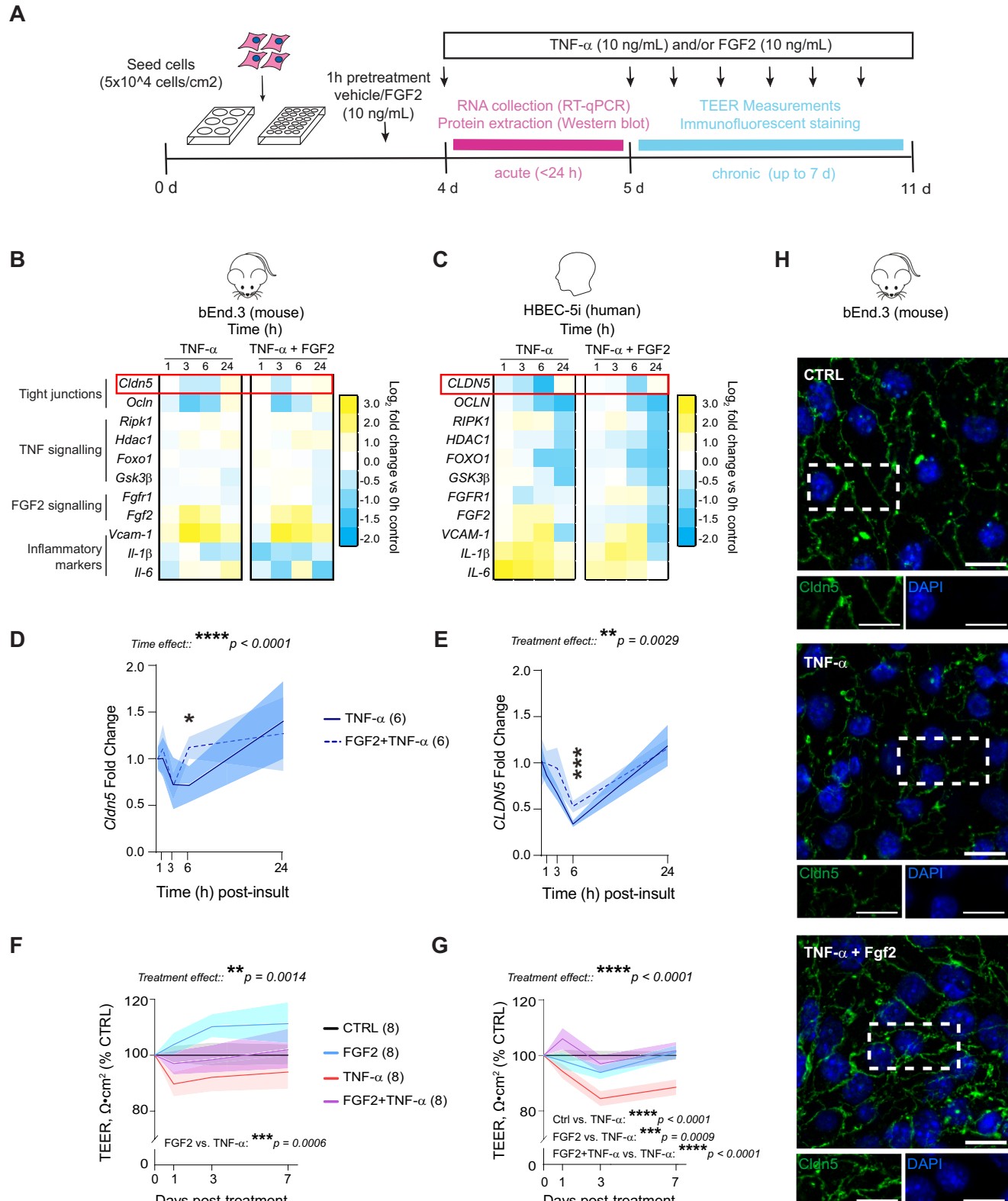

Fig. 6 | Treatment with Fgf2 reduces TNF-α-induced Cldn5 loss, endothelial cell signaling alterations, and barrier hyperpermeability. A, Experimental timeline for inflammatory insult with TNF-α and FGF2 co-treatment. Once confluent (4 days), HBEC-5i or bEnd.3 were pretreated with FGF2 or vehicle for one hour and then stimulated with TNF-α or vehicle for up to 7 days. FGF2 alters mouse (B) and human (C) endothelial transcription in response to acute TNF-α stimulation. FGF2 promotes faster restoration of TNF-α-induced Cldn5 loss in mouse (D) and human (E) endothelial cells. Chronic stimulation with TNF-α leads to a reduction in endothelial monolayer integrity measured by trans-endothelial electrical resistance (TEER) in mouse (F) and human (G) endothelial cells. FGF2 co-treatment preserves normal TEER despite TNF-α. H 7 days of TNF-α treatment promotes spikes and discontinuities in Cldn5 tight junction strands in bEnd.3, which is reversed by FGF2 treatment (scalebar = 20 μm). Data represent mean ± s.e.m., and each experiment was replicated at least twice on independent samples. Group comparisons were evaluated with two-way ANOVA followed by Bonferroni's post hoc tests; *p < 0.05, **p < 0.01, ***p < 0.001, ****p < 0.0001.

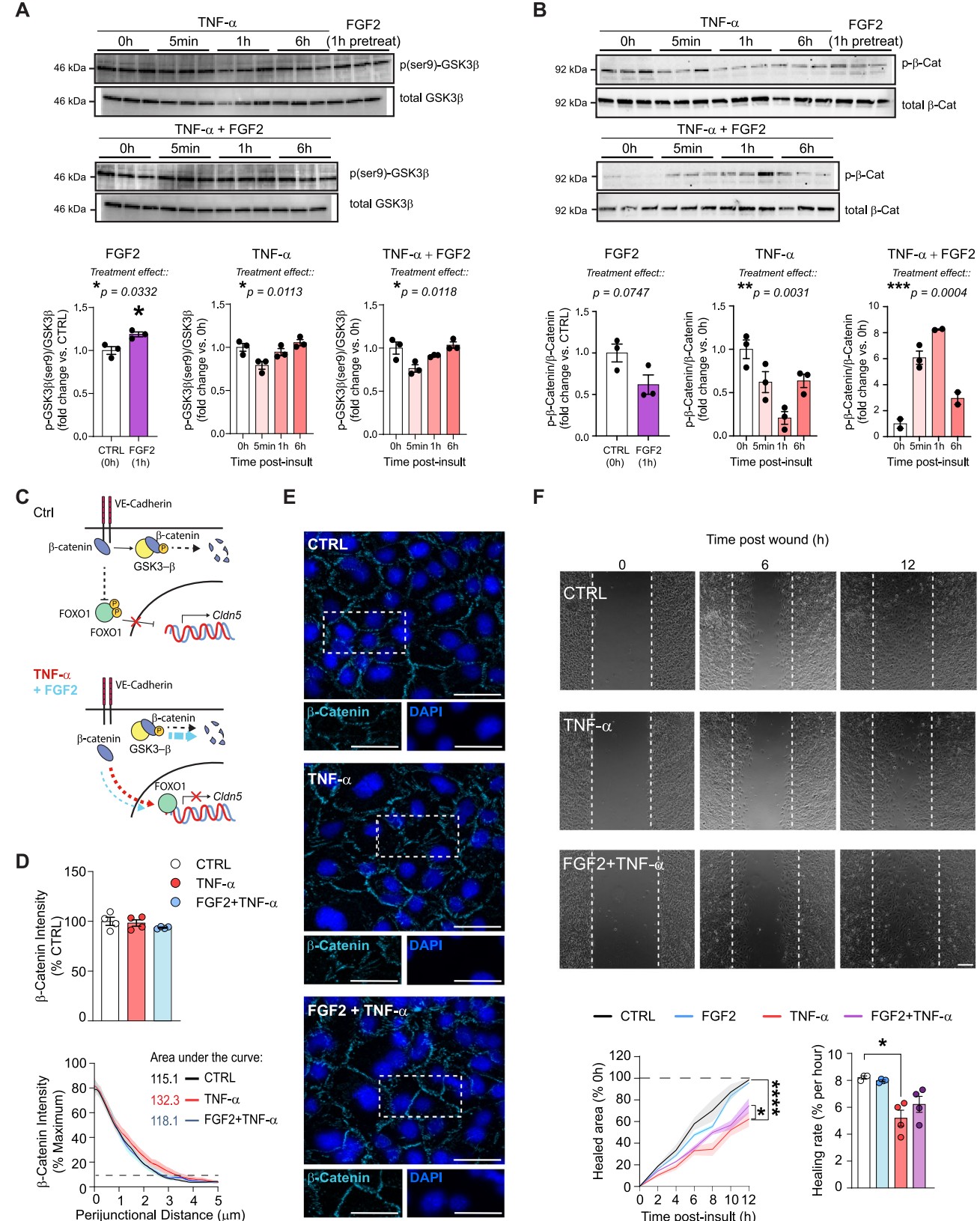

to the establishment of anxiety- and depression-like behaviors in female, but not male, mice. While direct comparisons of behavior and gene expression were not possible, we were still able to compare general patterns of change in BBB-related genes between CSDS males and SCVS females and found strong sex differences in the BBB response to stress under standard versus enriched environment. In

female mice, EE prevented SCVS-induced *Cldn5* loss in the PFC, but not in the NAc. SCVS reduces *Cldn5* expression in both PFC and NAc, but neurovascular disruption in the PFC alone is sufficient to promote depression-like behaviors in female mice[18]. Further, in contrast to males, EE did not promote an alternative transcriptional stress response in the female BBB but instead, stressed EE females

**Fig. 7 | Fgf2 induces GSK3β phosphorylation and prevents β-catenin dissociation from tight junctions. A** 1 h pretreatment with Fgf2 increases serine-9 phosphorylation of GSK3β in HBEC-5i when compared to no treatment (CTRL 0 h). TNF-α treatment induces rapid, transient dephosphorylation of GSK3β, but this effect is not reversed by Fgf2 coadministration. Each dot represents a replicate ($n = 3$). **B** 1 h Fgf2 pretreatment diminishes basal β-catenin phosphorylation when compared to no treatment (CTRL 0 h). Further, while TNF-α induces a rapid reduction in phosphorylated β-catenin, Fgf2 reverses this dynamic upon inflammatory activation ($n = 3$). **C** In health control endothelial cells (top), β-catenin interacts with VE-Cadherin at the cell membrane, and this complex inhibits *Cldn5* transcriptional suppression by FOXO1. Excess cytosolic β-catenin is phosphorylated by GSK3β, targeting it for degradation. When stimulated with TNFα, unbound β-catenin

complexes with FOXO1, leading to suppression of *Cldn5* expression (bottom, red arrow), while a small amount is targeted for degradation. Meanwhile, when FGF2 is co-administered with TNF-α (bottom, blue arrow), our results suggest that unbound β-catenin is strongly redirected toward GSK3β-mediated phosphorylation. **D** 30 min of TNF-α is sufficient to induce β-catenin distribution at tight junctions ($n = 4$ replicates) with representative images on the right (**E**) (scalebar = 20 μm). **F** Fgf2 attenuates TNF-α-induced reductions in the wound healing capacity of HBEC-5i ($n = 4$ replicates) (****$p < 0.0001$). Data represent mean ± s.e.m., and each experiment was replicated at least twice on independent samples. Group comparisons were evaluated with one or two-way ANOVA followed by Bonferroni's post hoc tests or two-tailed t-tests with Welch's correction when appropriate; *$p < 0.05$, **$p < 0.01$, ***$p < 0.001$, ****$p < 0.0001$.

---

maintained control-like gene expression. Clustering with PCA revealed that stressed males and females with EE access have distinct patterns of BBB transcriptional alterations. A possible explanation for sex differences in environment-BBB interactions during stress is that sex-specific steroid hormones are known to influence neurovascular unit function, potentially representing a controlling factor for determining the magnitude of impact from environmental change[59,96]. It will be important to investigate further how hormonal regulation, notably the estrus cycle and testosterone level, can affect behavioral responses, as sex hormones are potent modulators of BBB integrity[59,96]. To better discern the relationship between sex, environment, and stress, a stress paradigm known to consistently induce anxiety- and depression-like behaviors in animals of both sexes, like 21-day CVS[4], could be performed in future studies to enable direct comparisons of protective mechanisms.

The discovery of upregulated *Fgf2* gene and protein expression strictly in the male NAc after stress supports a sex-specific protective mechanism. Further, it suggests a common pathway associated with both EE and PE, implying that these conditions activate convergent biology in the brain to improve Cldn5 expression and BBB integrity. As mentioned above, Fgf2 has been shown to exert protective effects on endothelial cells and blood vessels, which could be related to the pro-resilient effects of stimulating environments. Also known as Fgf-basic or b-Fgf, Fgf2 belongs to the fibroblast growth factor family of proteins, which stimulate tissue growth and development in a variety of organ systems. It is produced in several variants, with the main secreted version being a low molecular weight 18 kDa isoform[97]. As opposed to other FGF family members, Fgf2 lacks a signal peptide and it is thus transported into the extracellular space by an ER/Golgi-independent mechanism, an unconventional secretory pathway[98]. A limitation of our study is that we combined immunofluorescence staining and confocal microscopy to quantify Fgf2 protein level in the mouse NAc and PFC and thus, we cannot precisely localize it and claim that it is secreted in the extracellular space. Future studies should consider taking advantage of super-resolution microscopy[99] which was previously used to map FGF receptor network[100]. Here, we chose to mimic EE- or PE-induced increase in *Fgf2* expression by upregulating expression of this gene with an AAV. An interesting option could have been to introduce a signal peptide in the viral construct to ensure Fgf2 secretion. However, it could impact normal secretion rhythm of this growth factor[101] and consistent Fgf2 overexpression increases seizure susceptibility in transgenic mice[102] which would create a confounding factor for behavioral studies. Finally, we cannot rule out compensatory changes following astrocyte-specific downregulation of *Fgf2* expression. Indeed, other factors can compensate for the lack of *Fgf2* leading to normal development and angiogenesis of *Fgf2−/−* and even double *Fgf1/2−/−* transgenic mice[36,103,104].

Fgf2 exerts its biological functions through four Fgf receptors (FGFRs), namely FGFR1, FGFR2, FGFR3, and FGFR4, with FGFR1 being the most highly expressed on endothelial cells[105]. Studies have linked Fgf2 signaling to a variety of beneficial effects, showing for example

that new blood vessels induced by Fgf2 show fewer fenestrations and less barrier leakage than vessels induced by vascular endothelial growth factors[44]. Inhibition of FGFR1 in endothelial cells leads to endothelial permeability and loss of tight junction protein expression[45]. In neurons, FGFR1 can form a complex with cannabinoid receptor 1 and activation of this receptor induces protein kinase C-dependent transactivation of FGFR1 through phosphorylation of key tyrosine residues of the receptor kinase domain[106]. We recently reported that astrocytic cannabinoid receptor 1 promotes resilience by dampening stress-induced BBB alterations and inflammation[107]. It will be interesting to investigate the potential relationship between FGFR1 and the cannabinoid system in astrocytes in the context of stress responses in future studies. Moreover, in both astrocytes and microglia, Fgf2 treatment is sufficient to diminish pro-inflammatory activation and cytokine release upon insult in vitro[108,109]. On the other hand, Fgf2 has been widely associated with anti-depressant and anxiolytic effects in rodents[43,46–49]. All this evidence suggests that increase of Fgf2 during stress in male EE and PE cohorts could be an adaptive response to protect against CSDS-induced tight junction loss and BBB permeability.

Next, we probed further into the role of Fgf2 at the BBB in the context of inflammatory damage to show how it may exert protective effects in endothelial cells in chronic social stress and MDD, both associated with elevated levels of circulating inflammatory cytokines[6,14,73,75]. As a simple model of stress-induced inflammation mouse and human brain endothelial cells were treated with TNF-α, a proinflammatory cytokine increased in the blood of humans with MDD and associated with circulating markers of vascular damage[60–62]. In addition, transcriptional pathways linked to TNF-α receptor signaling are upregulated in stress-susceptible mice[21]. Interestingly, vascular damage associated with TNF-α in a learned helplessness model of depression is gated by GSK3β which displays higher activity in learned helpless animals[22], corroborating in vitro findings that GSK3β mediates TNF-α-induced upregulation of leukocyte adhesion molecules on the brain endothelial cells[64]. Cldn5 plays an important functional role in the brain as the main tight junction protein regulating BBB permeability[110,111], and loss of this protein is observed in postmortem samples of humans with MDD[15,21,23]. Using our in vitro model, we found that Fgf2 protects against TNF-α-related loss of *Cldn5* expression in endothelial cells, suggesting a role as a protective agent at the BBB. In both mouse and human cells, Fgf2 attenuated *Cldn5* loss with a significant rescue apparent after 6 h of TNF-α treatment. Thus, rather than preventing initial *Cldn5* suppression, Fgf2 may engage slower-onset mechanisms[112] which result in an early termination of inflammatory signaling. Intriguingly, the degree of *Cldn5* loss was more pronounced in HBEC-5i versus bEnd.3 suggesting that the mouse cells were more resistant to inflammation. This corresponds to longer term effects on endothelial monolayer integrity with chronic treatment, where TNF-α induced a stronger loss of TEER in HBEC-5i than bEnd.3. However, in both species inflammatory damage does increase barrier permeability, and this is prevented by Fgf2 treatment in both cases.

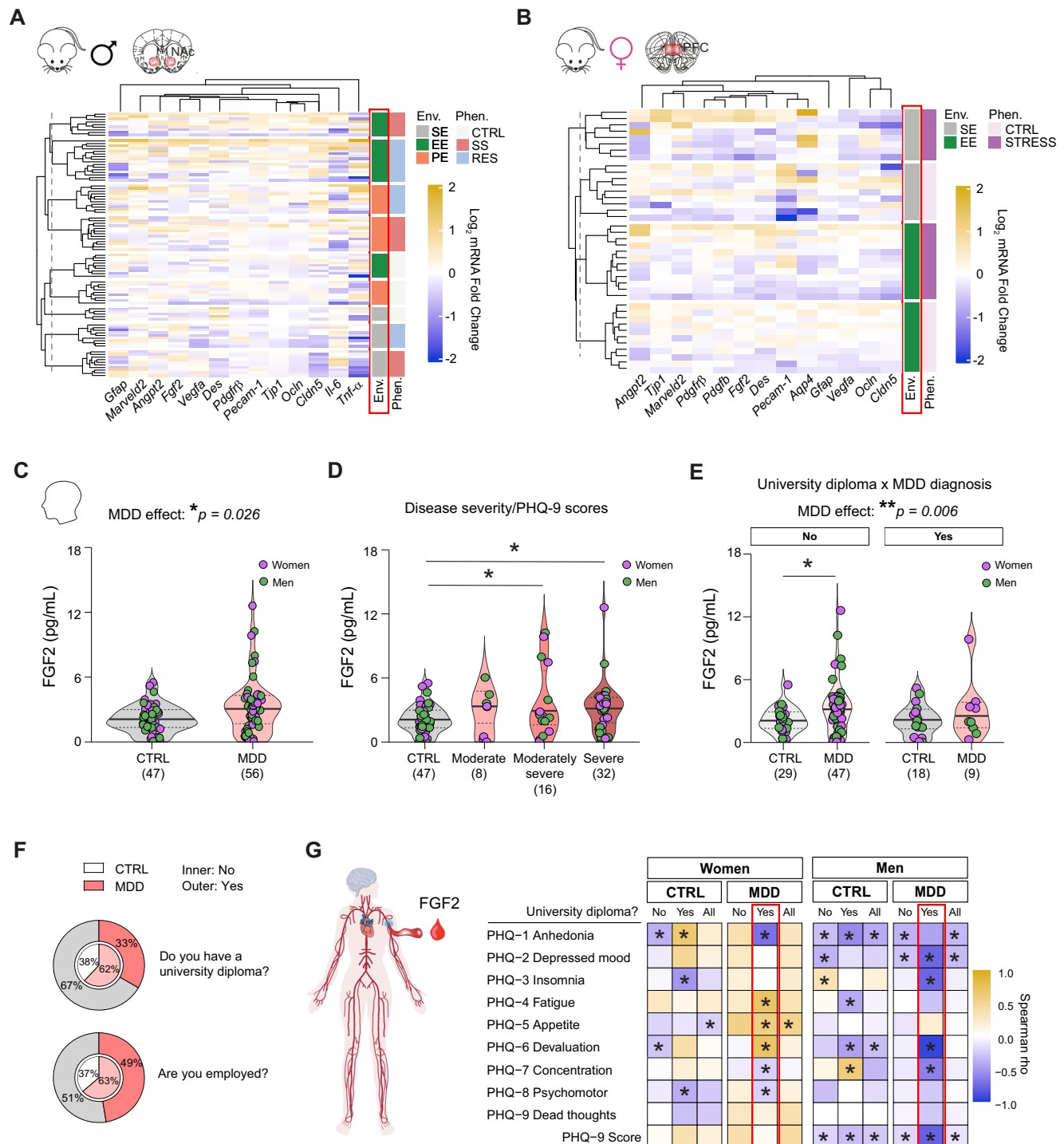

**Fig. 8 | The environment is a key variable determining BBB response to stress and FGF2 a biomarker of MDD severity and symptomatology.** Hierarchical clustering performed with Euclidean distances of blood-brain barrier (BBB)-related gene expression changes induced by stress exposure in the male nucleus accumbens (**A**, NAc) or female prefrontal cortex (**B**, PFC) in standard conditions (SE), with access to an enriched environment (EE), or voluntary physical exercise (PE). **C** Fibroblast growth factor 2 (FGF2) level in blood serum samples from men and women with a diagnosis of major depressive disorder (MDD) at various degrees of severity (**D**, minimal vs moderate severe: *$p$ = 0.0171; minimal vs severe:

*$p$ = 0.0452). **E** FGF2 blood level vs MDD diagnosis, sex, and education level (men with no diploma: *$p$ = 0.0172). **F** Proportion of individuals in each group with a university diploma (top) or employed (bottom). **G** Spearman correlation between circulating FGF2 and MDD symptoms according to the Patient Health Questionnaire (PHQ-9) item constructs. Data represent mean ± s.e.m., the number of individuals is indicated on graphs. Group comparisons were evaluated with two-way ANOVA followed by Bonferroni's post hoc tests or two-tailed Mann-Whitney U-test; *$p$ < 0.05, **$p$ < 0.01.

Furthermore, after 7 days of TNF-α treatment, bEnd.3 cells stained for Cldn5 show a large number of membrane spikes and discontinuities, which are typically associated with stress or strain on cell membranes[66]. Spike morphology results from dysregulation of Cldn5 at the tight junction interface, leading to increased paracellular leakage; further, these spikes have been identified as 'hot spots' of vesicular transport and potentially play a role in compromising selectivity of influx to the brain[66,113]. Reduced spike morphology in cells co-treated with TNF-α and Fgf2 therefore indicates a stabilization of Cldn5 at tight junctions, which could be related to prevention of BBB permeabilization following inflammatory challenge.

Fgf2 is known to interact with the Wnt/β-catenin system[114], which is crucial for the development of BBB properties and maintenance of TJ ultrastructure[115,116], suggesting a possible avenue for the protective effects of Fgf2 shown above. Dysregulation of β-catenin during CSDS is moreover a driver of epigenetic Cldn5 suppression[21], and we suspected Fgf2 may interfere with this process. TNF-α is a strong activator of GSK3β, which is thought to phosphorylate β-catenin, targeting it for degradation, during inflammation; conversely, Fgf2 stimulation activates signaling through Akt, a strong inhibitor of GSK3β[63,64,67]. In the absence of inflammation, we confirmed that Fgf2 increases deactivating serine-9 phosphorylation of GSK3β which corresponded with reduced β-catenin phosphorylation. Surprisingly, however, we found that β-catenin is modulated independently of GSK3β during the following TNF-α induction, where it is rapidly dephosphorylated despite activation of GSK3β. Furthermore, Fgf2 had no effect on TNF-α-induced GSK3β activation but completely reversed trends in β-catenin phosphorylation after TNF-α treatment. The observed drop in β-catenin phosphorylation as a result of TNF-α stimulation has also been reported in epithelial cells[117], but the biological significance of reversing this effect remains unclear. It is possible that this change influences β-catenin cell distribution, as TNF-α has been shown to induce β-catenin translocation to the nucleus where it suppresses Cldn5 expression, playing a role BBB disruption[70,117]. TNF-α-induced β-catenin accumulation in the cytoplasm could be related to inhibition of the proteasome by TNF-α, and thus less degradation of cytosolic β-catenin[118]. Further experiments will be needed to determine the precise mechanisms governing these drastic shifts in β-catenin phosphorylation and distribution during inflammation both with and without Fgf2.

In endothelial cells, β-catenin plays an important role in maintaining cell-cell adhesions, binding to VE-cadherin and linking it to the actin cytoskeleton in a structure with implications for both tight junction stability and cell motility[119]. We show that just 30 min of TNF-α treatment results in diffusion of β-catenin staining from cell-cell contacts, suggesting this as an early step in inflammation-related disruption of the junctional structure. Distension of cadherin complexes is potentially indicative of disrupted interaction between tight junction architecture and the actin cytoskeleton, an effect which could ultimately be related to the development of Cldn5 spikes and barrier hyperpermeability after chronic inflammatory damage[70]. Further evidence of dysfunction in membrane-cytoskeleton communication following TNF-α stimulation is a strong reduction of wound healing rate indicative of a decline in cell motility. β-catenin strongly regulates vascular cell growth and destabilization of this protein by TNF-α could be related to compromised repair mechanisms[72]. These connections are interesting but future experiments will be needed to confirm effects of TNF-α on actin dynamics at tight junctions as well as investigate causal relationships between this and changes to Cldn5 distribution and expression in the context of chronic stress exposure or mood disorders. Nevertheless, our results support TNF-α-induced dysfunction at sites of cell adhesion, and the fact the Fgf2 restores control-like β-catenin labeling as well as slightly rescuing wound healing rate demonstrates a stabilizing effect in the face of inflammatory damage.

Dysregulation of FGF signaling has been implicated in mood disorders, including MDD, and measurement of this growth factor as a brain and blood biomarker of psychiatric conditions has been considered yielding inconsistent results. Indeed, lower levels of FGF2 were reported in the dorsolateral PFC and anterior cingulate cortex[52] as well as hippocampus[120] of postmortem samples from depressed individuals; however, no change was noted by others[76]. *FGF2* is mostly expressed by astrocytes in the brain, but also other cell types[34,35]. Recent development and refining of single-cell sequencing techniques shed light on the brain cellular heterogeneity including for glial cells and components of the neurovascular unit[7]. Taking advantage of these technologies may be useful to resolve these discrepancies in future work. Contradictory findings have also been reported for FGF2 blood serum or plasma level for individuals with psychiatric conditions when compared to matched healthy controls with decreased[121], increased[78–80,122], or no difference[77]. This could be due to heterogeneity of clinical aspects in the human cohorts, such as comorbidities, treatments, symptoms, and lived experience. Unfortunately, confounding sources are an inherent limitation of human biomarker studies. As reported here, socioeconomic factors may have an impact and should be carefully considered. Detection of FGF2 itself in the plasma/serum is challenging due to its affinity to heparin[123] and rapid turnover, therefore, highly sensitive immunoassays, like the one used here, are strongly recommended. To conclude, we investigated, here, BBB-related cellular and molecular mechanisms underlying resilience to stress and their promotion by an enriched environment, mimicking to some extent high socioeconomic status in humans, or physical exercise as an intervention to favor neurovascular health. We identified the growth factor Fgf2 as a promising target to protect the BBB when facing social adversity and argue that it could represent an interesting biomarker to move towards personalized medicine and tailored treatment in the context of mood disorders.

## Methods

### Animals

Male and female C57BL/6 mice aged 8 weeks of age at arrival (Charles River Laboratories, Québec, Canada) were used for all experiments. Retired male CD-1 breeders were used as resident aggressors (AGG) for social defeat and social interaction tests. All mice were group housed in 27 × 21 × 14 cm polypropylene cages upon their arrival and left undisturbed for at least three days prior to experimentation. Mice were maintained on a 12-h light–dark cycle (lights on from 0700 to 1900 h) with constant temperature, humidity (22 °C, 63%) and free access to water and food (Teklad Irradiated Laboratory Animal Diet, Madison, USA). All experimental procedures were approved by the Animal Care and Use Committee of Université Laval (2022-1061-1) and met the guidelines set out by the Canadian Council on Animal Care.

### Housing conditions

Control and stressed EE mice were housed in a standard cage supplemented with a plastic house, nesting material, and a small plastic toy. When animals were moved between cages during CSDS, they maintained their original enrichment materials. Control and stress animals in the PE cohorts were housed in standard cages and habituated with battery-powered, wireless running wheels (Med Associates) for 5 days prior to the beginning of CSDS. This habituation period is based on previous reports as well as our observations (not shown) that running activity per day reaches a plateau after five days. Each animal was assigned a wheel which it was kept with throughout cage changes during the 10 d CSDS protocol. Data was collected and exported at 1-min intervals using Wheel Manager software (Med Associates).

### Chronic social defeat stress (CSDS)

Male C57/Bl6 mice underwent CSDS as previously described[41]. AGG mice underwent 3 days of screening for aggression profile and were conditioned in social defeat cages separated halfway by a clear,

perforated divider for 24 h prior to experiments. Experimental mice were subject to physical interaction with a novel CD-1 for 5 min a day over 10 consecutive days and subsequently housed in defeat cages opposite the CD-1 with the divider preventing physical altercation but allowing sensory contact. Interactions were stopped before the 5-min period elapsed if attacks were repeated and severe, or if wounding occurred. Unstressed controls were co-housed in social defeat cages on each side of a divider and were moved every other day. After the last bout of interaction, the experimental mice were single-housed in standard cages for 24 h before undergoing a social interaction (SI) test, and tissue was collected 24 h after that (Fig. 1A).

### Chronic variable stress (CVS)

Female C57/Bl6 mice were housed in groups of four in standard cages for 6 d CVS protocol[18]. Briefly, stressed mice were subject to three different alternating stressors, one per day, in the following order: 100 random mild foot shocks (0.45 mA) for 1 h, tail suspension for 1 h, and tube restraint within home cage for 1 h. Unstressed controls were handled every day. After the last stressor, mice were single-housed for behavioral testing and tissue was collected 24 h after the last test (Fig. 2A).

### Social interaction (SI) test

SI tests to assess social preference were performed under red light conditions[15,41]. Mice were placed in an open field arena with a small wire cage at one end for 150 s. Mice were then removed and the arena was cleaned, a CD-1 (AGG) was placed in the wire cage, and experimental mice were again allowed 150 s to freely explore the arena. Behavior in presence and absence of social target was tracked with AnyMaze software. Interaction zone (IZ) is defined as the area around the mesh cage. SI ratio was calculated by dividing the time in interaction zone in presence vs. absence of AGG. Mice with SI < 1 were classified as stress-susceptible (SS), while SI = 1 or >1 were resilient (RES).

### Elevated plus maze (EPM)

The EPM apparatus is a cross-shaped plexiglass arena with 4 arms (12 cm width × 50 cm length) 1 m above ground level, where two arms had tall black walls (closed arms) and two were unprotected (open arms). Under red light, mice were placed in the middle of the maze and allowed to explore for 300 s. Behavior was automatically tracked (AnyMaze 6.1, Stoelting Co.). Time in closed arms is taken as a measure of anxiety-like behavior.

### Sucrose preference test

Water bottles in standard cages were replaced with two 50 mL conical tubes containing water for a 48-h habituation. Next, water from one of the tubes was replaced with 1% sucrose and mice were allowed to drink ad libitum. Tubes were switched after 24 h to account for place bias, and weights were recorded at 0 h, 24 h, and 48 h. Sucrose preference after 24 h was calculated by dividing weight of sucrose consumed by the total weight of liquid.

### Forced swim test (FST)

Forced swim is used to evaluate learned helplessness as a measure of depression. Mice were placed in a 4 L glass beaker filled with 3 L lukewarm water under bright light for 360 s. Video of each session was manually evaluated for time spent immobile, defined as no movement or small hind-leg gestures needed to stay afloat, by blinded observers.

### Viral experiments

At 5 weeks of age, mice were injected either AAV5-gfaABC1D-GFP, AAV5-gfaABC1D-m*Fgf2*-P2A-GFP, AAV5-gfaABC1D-shRNAmir(-scramble)-GFP, or AAV5-gfaABC1D-shRNAmir(m*Fgf2*)-GFP (Sigma Gene) viruses via stereotaxic injection. All surgeries were performed under aseptic conditions using anesthetic as previously described[15,18].

Mice were anesthetised with isoflurane (3-4% in $O_2$ for induction and 1–1.5% in $O_2$ for maintenance at 1 L/min flow rate) and positioned in a small stereotaxic instrument (Harvard Apparatus). Skull was exposed, and 0.5 ul (0.2 ul mere viral solution +0.3 ul saline) was bilaterally infused in the NAc (bregma coordinates: anteroposterior +1.6 mm; mediolateral ±1.5 mm; dorsoventral −4.4 mm), at a rate of 0.1 ul/min. For FGF2 staining to confirm FGF2 over-expression, mice were allowed to recover for 4 weeks, then brains were collected and flash frozen with isopentane on dry ice and store in −80 ℃ until use. For viral injection + CSDS experiment, mice were allowed to recover for ~4 weeks, then went under CSDS as described above. Viral groups were divided between unstressed controls group and stressed group ($n = 6$ and 12, respectively). SI was performed after 10 days CSDS, then the following day 24 h later, mice were sacrificed by rapid decapitation, and their brains were collected via NAc bilateral punches of 2 mm from 1 mm slices.

### Cell culture

The human brain microvascular endothelial cell line HBEC-5i (ATCC CRL-3245, male donor according to https://www.cellosaurus.org/CVCL_4D10) and the mouse brain endothelial cell line bEnd.3 (ATCC CRL-2299) were subcultured and stored in banks at −150 ℃. Cells were thawed as needed and cultured in DMEM/F12 supplemented with 10% fetal bovine serum, 25 ug/mL gentamicin (Gibco), and 1X endothelial cell growth supplement (ScienCell). Culture surfaces were precoated with 0.1% gelatin and cells were passaged when confluent (3–5 days) by washing with PBS, detaching with TrpLE Dissociation Reagent (Gibco), and seeding on gelatin coated flasks at desired concentration. Cells were used for experiments between passages 3 and 7, seeded at a density of $5 \times 10^4$ cells/cm$^2$. For immunofluorescence, cells were grown on gelatin-coated 12 mm glass coverslips (Marienfeld), which were previously hydrophilized by 10 min treatment with 0.1 M hydrochloric acid and sterilized for 10 min with 100% ethanol.

### TNF-α and Fgf2 treatment

Human recombinant TNF-α and FGF2 (Gibco) were dissolved in sterile water per manufacturer's instructions and stored at −20 ℃. Experiments were performed using HBEC-5i and bEnd.3 after 4–5 days in vitro. Cells were pretreated for 1 h with 10 ng/mL human recombinant FGF-2 (Gibco) or vehicle (sterile water). At the start of treatment, existing medium was completely aspirated and replaced with HBEC-5i cell culture medium containing either sterile water (vehicle), 10 ng/mL FGF2 + vehicle, vehicle +10 ng/mL TNF-α, or 10 ng/mL FGF2 + 10 ng/mL TNF-α. Acute inflammation studies looked at the response to a single stimulus up to 24 h, while chronic inflammation was assessed by replacing the medium each day with fresh medium containing TNF-α and/or FGF2 over a period of up to 7 days.

### Gene expression analysis

Mice were anesthetized by decapitation and brains were rapidly removed. 2.0 mm punches were taken from NAc and PFC in both hemispheres and frozen at -80 ℃ until use. HBEC-5i and bEnd.3 in a 6-well plate were pretreated for 1 h with Fgf2 or sterile water (vehicle) and then stimulated with TNF-α or TNF-α + FGF2 for 0 h (control), 1 h, 3 h, 6 h, or 24 h (3 wells/condition/timepoint). RNA was extracted from brain punches as well as HBEC-5i and bEnd.3 cells in 6-well plates using TRIzol (Invitrogen) homogenization and phase separation with chloroform. The clear RNA phase was processed further with the Pure Link RNA MINI Kit (Life Technologies) and assessed for purity and concentration with NanoDrop (Thermo Fisher Scientific). Complementary DNA (cDNA) was obtained with a reverse transcriptase reaction using Maxima-H-minus cDNA synthesis kit (Fisher Scientific). For qPRC reactions, each well of a 384-well plate contained 3 μL of sample cDNA, 1 μL qPCR primer (see Supp. Table 1 for primer list), 5 μL Power up SYBR green (Fisher Scientific), and 1 μL distilled $H_2O$. In a

thermocycler, samples were heated to 95 °C followed by 40 cycles of 95 °C for 15 s, 60 °C for 33 s and 72 °C for 33 s. Ct values were converted to normalized expression using the $2^{-\Delta\Delta Ct}$ method[124] with *Gapdh* as the reference gene.

## Immunofluorescent staining

Whole brains of mice after rapid decapitation were flash frozen with isopentane on dry ice and stored at −80 °C until use. Frozen brains were embedded in OCT Compound (Thermo Fisher Scientific) and slices from PFC and NAc were collected using a cryostat (Leica) at 20 μm thickness or 30 μm for AAV5-Fgf2-related experiments. Brain slices and cells cultured on glass coverslips were post-fixed for 10 min in ice-cold methanol. After 3× 5 min wash with PBS brain slices were incubated for 2 h in blocking solution (1% bovine serum albumin, 4% normal donkey serum, and 0.03% Triton X-100 in PBS) before overnight incubation with primary antibodies in blocking solution (see Supp. Table 2 for antibodies and dilutions). Samples were then washed 3 × 5 min with PBS and incubated with fluorophore-conjugated secondary antibodies in PBS, then washed again and stained with DAPI to visualize nuclei. Coverslips were mounted on slides using Prolong Diamond Antifade Mountant (Invitrogen). Z-stack images of the NAc and PFC were taken at 20× ($Z = 10$ μm) and 40× ($Z = 3$ μm or 4.8 μm for AAV5-Fgf2-related experiments) on an epifluorescence microscope (Carl Zeiss).

## Fluorescent image analysis

Analysis of brain tissue images was automatically performed in batches using Fiji ImageJ software[125]. For each channel, maximum intensity projection was performed and resulting images were processed with rolling ball background subtraction followed by noise removal of bright artefacts less than 2 μm. In CD31 channel only, continuity of blood vessels was ensured by 2D Gaussian blur, $\sigma = 2$ μm. Processed images were binarized using automatic thresholding algorithms to measure positive staining area. β-catenin distribution at tight junctions in HBEC-5i cells was assessed by a blinded experimenter who sampled one tight junction from each quadrant of each image for a total of 48 TJs per condition. β-catenin distribution was assessed by measuring fluorescence intensity across a 10 μm line drawn perpendicular to the junction plane.

## Cell viability assay

3-(4,5-dimethyl-2-thiazolyl)-2,5-diphenyl-2H-tetrazolium (MTT) (Millipore Sigma) is converted to water-insoluble product formazan by reduction at mitochondrial complex II of the electron transport chain. This reaction can therefore act as a proxy for mitochondrial respiration and cell viability[126]. Briefly, cells in a 24-well plate were treated with 500 μM MTT and incubated (37 °C and 5% $CO_2$) for 2 h. Media was removed, and formazan crystals were dissolved in 500 μL dimethyl sulfoxide for absorbance readings at 570 nm using a spectrophotometer. Viability is calculated as a percent of the control reading.

## Trans-endothelial electrical resistance (TEER)

For TEER studies, HBEC-5i were seeded on transwell polycarbonate culture inserts (Millicell) with 12 mm diameter and 3 μm pore size. TEER measurements were taken using the Millicell® ERS-2 Electrical Resistance System. Gelatin-coated insert with no cells was used as a blank. Electrodes were habituated in complete growth media at room temp for 10 min before reading resistance across cell monolayers. TEER was calculated as resistance of sample minus resistance of blank, multiplied by membrane surface area (0.6 cm²). TEER measurements were normalized to baseline for each well and are presented as percentage of control; raw TEER values are available in Supp. Fig. 7.

## Protein extraction and western blot

HBEC-5i and bEnd.3 in a 6-well plate were pretreated for 1 h with Fgf2 or sterile water (vehicle) and then stimulated with TNF-α or TNF-α +

FGF2 for 0 h (control), 5 min, 1 h, or 6 h (3 wells/condition/timepoint). Protein was extracted at desired timepoints by washing with ice-cold PBS and then lysing cells with 200 μL cell lysis buffer (Cell Signaling Cat. No. 9803) supplemented with protease inhibitor cocktail (Cell Signaling Cat. No. 5871). Samples were sonicated in ice-cold water, centrifuged for 10 min ($17,949 \times g$, 4 °C), and supernatant transferred to a new tube. Protein level was quantified using the Pierce™ BCA Protein Assay Kit (ThermoFisher Cat. No. 23250) per manufacturers instructions. Samples were diluted 1:10 and absorbance values were read on a spectrophotometer at 562 nm. Protein levels were calculated from standard curve. Samples were prepared for gel electrophoresis by adding a volume containing 20 μg protein to 3 μL 1 M DTT and 7.5 μL 4X Laemmli sample buffer (BioRad Cat. No. 1610747) and topping up to 30 μL with deionized water. Samples and protein ladder (10−250 kDa, Cell Signaling Cat. No. 74124) were pipetted into wells of a 18-well Criterion™ TGX Stain-Free™ pre-cast gel (4−15%, BioRad Cat. No. 5678084) and separated by SDS-PAGE. Protein was transferred to a polyvinylidene difluoride membrane with 60 min of 90 V current in transfer buffer (25 mM Tris base, 192 mM glycine, 20% methanol in deionized water). Membranes were blocked in Tris-buffered Saline (TBS) supplemented with 0.1% Tween (TBST) and 0.5% BSA and incubated overnight with primary antibodies in blocking solution at 4 °C (see Supp. Table 2 for antibodies and dilutions). Membranes were washed 3 × 10 min in TBST and incubated 1 h with horseradish-peroxidase (HRP)-conjugated secondary antibodies in TBST. Membranes were washed again and signals on blots were revealed by enhanced chemiluminescence (ECL) reagents (BioRad Cat. No. 1705060) in a ChemiDoc imaging system (BioRad). Band intensity was estimated in Fiji ImageJ by removing background separately for each lane and measuring volume of the peak signal[127,128]. Phosphorylated GSK3β and β-Catenin levels were normalized to total GSK3β and β-Catenin. Total protein was measured with Stain-Free™ imaging technology and used as a loading control for Cldn5 levels.

## Scratch wound assay

HBEC-5i in 24-well plates were pre-treated with Fgf2 for 1 h and subsequently stimulated with Fgf2, TNF-α, Fgf2 + TNF-α, or sterile H2O as vehicle (CTRL) for 24 h prior to wound induction. Scratch wound was induced using a sterile 200 μL pipette tip aligned with a custom 3D-printed plastic template to ensure all scratch sizes were equal, following the recommendations of previous publications[129]. Wounded cells were washed twice with DMEM/F12 and subsequently incubated with Fgf2, TNF-α, Fgf2 + TNF-α, or sterile H2O as vehicle. Images were taken at 5× on the brightfield setting of an epifluorescence microscope at 2 h intervals beginning immediately after scratch and continuing until 12 h. Wound size was manually evaluated using the Wound Healing Size Tool plugin for Fiji[129].

## Human serum sample collection

All human blood samples were provided by Signature Bank from the Centre de recherche de l'Institut universitaire en santé mentale de Montréal (CR-IUSMM) under approval of the institution's Ethics Committee. Samples from volunteers with major depressive disorder were collected at the emergency room of the Institut universitaire en santé mentale de Montréal of CIUSSS de l'Est-de-Montreal, and samples from healthy volunteers at the CR-IUSMM. All donors provided informed consent and signed a 7-page document detailing the goals of the Signature Bank, participants involvement (questionnaires and tissue sampling), advantages vs risks, compensation, confidentiality measures, rights as participant and contact information. Subjects with known history of drug abuse were excluded. Demographic characteristics associated with each sample are listed in Supp. Table 3 (sex was self-reported). Depressive behaviors were assessed by the Patient Health Questionnaire (PHQ-9), which scores each of the nine Diagnostic and Statistical Manual of Mental Disorders (DSM) IV criteria[130].

All experiments were performed under the approval of Université Laval and CERVO Brain Research Center Ethics Committee *Neurosciences et santé mentale (Project #2019-1540)*.

### Enzyme-linked immunosorbent assay (ELISA)

Serum levels of FGF2 were assessed with the Human FGF basic/FGF2/bFGF Quantikine® high sensitivity ELISA kit from R&D systems (Cat. No. HSFB00D, sensitivity 0.07 pg/mL, assay range 0.3-20 pg/mL), following manufacturers instructions. Serum samples were diluted 1:2 and absorbance read at 490 nm nm with wavelength correction at 690 nm on the microplate/spectrophotometer Agilent Biotek Epoch 2. Concentrations were calculated from a 4PL standard curve using the R package drc.

### Statistical analysis

Statistical comparisons were performed using GraphPad Prism 9 software. Each dataset was tested for normality (Shapiro-Wilk test, alpha = 0.05) and outliers (Grubb's test, alpha = 0.05). Animals identified as outliers in two or more distinct behavioral measures were removed from further analysis. Two-group comparisons were performed using two-tailed unpaired Welch's t-test (normal distribution) or Mann-Whitney U test (non-gaussian distribution). Multiple group comparisons were assessed with one- and two-way analysis of variance (ANOVA) or multiple permutations (non-equal variances) followed by Bonferroni post-hoc testing (normal distribution) or Kruskal-Wallis test with Dunn's post-hoc test (non-gaussian distribution). Principal component analysis (PCA) was performed using the R software, package FactoMineR, and missing values were imputed with missMDA.

### Reporting summary

Further information on research design is available in the Nature Portfolio Reporting Summary linked to this article.

## Data availability

All data supporting the findings of this study are available within the paper and Supplementary Information. Source data are provided with this paper.

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

## Acknowledgements

This research was supported by the Canadian Institutes for Health Research (CIHR, Project Grant #427011 and #495641 to C.M.), Fonds de recherche du Quebec—Santé (FRQS, Junior 2 salary award to C.M.) and C.M. Sentinel North Research Chair funded by Canada First Research Excellence Fund. S.E.J.P., J.L.S., A.Cadoret, A.Collignon, L.B.B., B.D., L.M.B., L.D.A., K.A.D. are supported by scholarships from CIHR, Réseau québécois sur le suicide, les troubles de l'humeur et les troubles associés (RQSHA), Natural Sciences and Engineering Research Council of Canada (NSERC), NeuroQuebec and FRQS. The Signature Consortium acknowledges contributions to the Biobank Signature of the CR-IUSMM (www.banquesignature.ca). The Biobank Signature received funding from the Fondation de l'Institut Universitaire de Santé Mentale de Montréal, *Bell cause pour la cause,* and the RQSHA. The authors would like to sincerely thank Dr. Jack McGugan from the Department of Anesthesiology and Perioperative Medicine at Queen's University, who performed the 3D printing of our scratch wound template.

## Author contributions

S.E.J.P. and C.M. designed the research. S.E.J.P., J.L.S., A.Cadoret, A.Collignon, L.B.B., B.D., L.M.B., E.R., F.C.R., L.D.A., K.A.D., and M.L. performed the research, including behavioral experiments, molecular, biochemical, and morphological analysis. The Signature Consortium contributed the human blood samples and related demographic and sociodemographic data. S.E.J.P., J.L.S. and C.M. analyzed the data and wrote the manuscript, which was edited by all authors.

## Competing interests

The authors declare no competing interests.

## Additional information

## Signature Consortium

Philippe Beauchamp-Kerr[2], Felix-Antoine Berube[2], Janick Boissonneault[2], Francois Borgeat[2], Lionel Cailhol[2], Pierre David[2], Simon Ducharme[2], Alexandre Dumais[2], Helen Findlay[2], Stéphane Guay[2], Steve Geoffrion[2], Charles-Edouard Giguere[2], Roger Godbout[2], Alexandre Hudon[2], Robert-Paul Juster[2], Real Labelle[2], Marc Lavoie[2], Myriam Lemyre[2], Alain Lesage[2], Cécile Le Page[2], Olivier Lipp[2], Sonia Lupien[2], Jean-Pierre Melun[2], Marie-France Marin[2], Carolle Marullo[2], Francois Noel[2], Jean-Francois Pelletier[2], Vincent Tascherau-Dumouchel[2], Pierrich Plusquellec[2], Stephane Potvin[2], Ahmed-Jérome Romain[2], Marc Sasseville[2], Daniel St-Laurent[2], Manuel Serrano[2], Emmanuel Stip[2], Christo Todorov[2], Valerie Tourjman[2], Samir Taga[2], Claudia Trudel-Fitzgerald[2], Martha Francoise Ulysse[2] & Andreas Ziegenhorn[2]

[2]Institut Universitaire en Santé Mentale de Montréal, Centre Intégré Universitaire de Santé et Service Sociaux Est, Montreal, QC, Canada.

