## [Transparent Peer Review file · Nature Communications]

Environmental enrichment and physical exercise prevent stress-induced social avoidance and blood-brain barrier alterations via Fgf2.

Corresponding Author: Professor Caroline Menard

Version 0:

Reviewer comments:

Reviewer #1

(Remarks to the Author)

This is an interesting study that provide new mechanistic insights into the FGF2-mediated protective effect of BBB by preventing Cldn5 loss and mitigating TNFa-mediated neuroinflammation and vascular inflammation. Exogenous delivery of AAV-FGF-2 improves depression-like behavior, zindlujcing social avoidance. Finally, they correlated the circulating FGF-2 levels with MDD in humans and reported a significant difference between men and women and the degree of MDD severity. In general, these findings are interesting and clinically relevant. I have major concerns regarding the FGF-2 biology in relation to MDD.

1. It seems to me that three growth factors, including VEGFA, PDGF, and FGF2, are somehow predetermined signaling molecules owing to their previously known involvements in BBB. This is a biased selection of growth factors. The authors should have deployed an unbiased approach to identify growth factors or cytokines that play key roles in their animal models, but not too much affected by what have been known.
2. The FGF family consists of more 20 members and some of the these members have distinct expression patterns in the brain. I am uncertain if the FGF-2 is the only FGF expressed in PFC and NAc. Given the key sex difference in their stress models, it is hard to believe that FGF-2 would execute the protective function in both cases. Other FGF members would likely to play a crucial role. It would have been really valuable if the authors looked into the differential expression profiling of growth factors in PFC and NAc areas as the initial step for the project.
3. FGF-2 is not a classical secretory protein. It lacks a signal peptide and remains as an intracellularly cytoplasmic protein. While the authors detected transcription expression of Fgf2 mRNA in their models, it does not mean export of FGF-2 to the extracellular space.
4. In the AVV system for the enforced expression of FGF-2, the authors should have introduced a signal peptide to ensure secretion of FGF-2 and have replaced the cysteine residues with other animo acids. Otherwise, the structure of FGF-2 would be changed by forming disulfide bonds.
5. Detection of FGF-2 in the plasma/serum usually represents noisy backgrounds. In this case, subtle differences between 5-7.5 pg/ml should not be considered as real differences. Owing to its extremely high affinity to heparin, the circulating FGF-2 would not be detectable because the FGF-2 molecules will be trapped by interaction with heparin sulfate proteoglycans in the extracellular matrix and on the cell surface. In fact, injection of FGF2 into the blood stream results in a rapid turnover.

In summary, while approaching the stress disorders using experimental models is extremely important and these authors have tremendous expertise in this field, identification of signaling molecules and mechanistic insights into their functions remain very vague. Much more rigors are needed to justify their conclusions.

Reviewer #2

(Remarks to the Author)

The manuscript, "Environmental enrichment and physical exercise prevent stress-induced social avoidance and blood-brain barrier alterations via Fgf2" by S.E.J. Paton, J.L. Solano, A. Cadoret, A. Collington, L. Bandeira Binder, E. Richer, F. Coulombe-Rozon, L. Dion-Albert, K.A. Dudek, Signature Consortium, M. Lebel & C. Ménard is a potentially exciting examination of the sex-specific protective effects of Fgf2 in attenuating stress-induced loss of the endothelial tight junction

protein, Claudin-5, decreasing anxiety- and depressive-like behaviours and protection against an inflammatory challenge. The authors suggest that the protective effects of an enriched environment and voluntary physical exercise on psychological stress are, in part, mediated through an increase in the astrocyte-derived growth factor, Fgf2, which has been previously reported to have anxiolytic and antidepressant effects. Furthermore, the authors demonstrate that this increase in Fgf2 is related to BBB changes, which is known to be impaired in individuals with MDD.

Changes in the BBB in MDD are clearly understudied and are likely important to both the etiology and treatment of MDD. In theory, this manuscript could provide a novel step forward in understanding precisely how environmental factors may protect against many stress-induced effects, particularly through changes in BBB function. However, some conclusions made in the manuscript are an overreach for the data presented. In addition, there are important technical limitations that need to be remediated.

Major issues:

-The authors correctly note that sex differences in MDD are marked and it is appreciated that they investigate these given the lack of studies in females in preclinical work. However, in this case, they chose to employ two different stress models, with males exposed to CSDS and females to SCVS, and then go on to make conclusions about sex differences in response to EE. CSDS and SCVS models are not interchangeable as they model different types of stress-induced depressive phenotypes. EE may have different effects in each of these models, regardless of sex and therefore conclusions cannot be made as to sex differences. Importantly, the authors again correctly note that CSDS is not easily transferable to females, however SCVS can be employed in males and therefore it is not clear why they do not include males in their SCVS study to make direct conclusions about sex differences in EE effectiveness.

-In several statements (including the title), the authors conclude that "Environmental enrichment and physical exercise prevent stress-induced social avoidance and blood-brain barrier alterations via Fgf2", however they do not test whether blocking FGF2 blocks the effects of EE, PE and the BBB alterations. They show that FGF2 coincidentally goes up with PE and EE and they show that increasing FGF2 in the male NAc "mimics" the effects of PE and EE on social avoidance, however they do not show that FGF2 is causally involved in the effect of PE and EE.

-Why does FGF2 prevent social avoidance in CSDS but not SI? EE alone changes SI and social avoidance, so is FGF2 really mimicking EE? Importantly, the levels of FGF2 reported with their AAV are quite high and therefore it is striking that the behavioural effects are limited. Are changes in social avoidance behaviour alone meaningful? Is this potentially an off-target effect? For example, does increased FGF2 change activity levels in general? FGF2 KOs show decreased centre time in the open field, so is this simply a reduction in anxiety/increased exploration?

-Figure captions and methods for calculating scores need to be expanded. For example, the authors report that they use control data from past studies to compare with EE and PE. Were any controls included in the current work? How do they ensure that behavioural baselines in controls are consistent across studies?

-It appears as though the authors used the DFB50 kit from R&D to measure FGF2 levels and not the high sensitivity (HS) kit. This kit is normed for clinical oncology samples with high levels of FGF2 and is sensitive from 10-640 pg/ml and is not sensitive for low levels of FGF2 typically seen in non-oncology samples. The HS kit is typically used for samples with low (or non-pathological) FGF2 levels as its sensitivity is between 0.3-20 pg/ml. Levels of FGF2 in Figure 7 are clearly below the sensitive range for the kit employed and therefore results may be confounded by a floor effect and conclusions cannot be drawn. Samples should be re-processed with the correct assay in order to have enough sensitivity to detect effects. Importantly, this may also help to resolve the inconsistencies in previous findings that the authors note.

-The data with respect to diploma and second language seems slightly out of left field. While social determinants of mental health are indeed critical, it is not clear what these two variables mean in this context?

-It might be interesting to correlate FGF2 levels to symptom clusters rather than these social determinants given previous work with FGF2 relating to specific phenotypes in humans.

Minor issues:

-How were the doses and timing of FGF2 and TNF- α chosen?

-Figure 6B- it appears as though p-B-catenin is increased at 0 hours with TNF α and not with TNF α + FGF2? Also, what does CTRL refer to?

-The authors refer to the PFC generally but may want to specify as it looks as though they are investigating the mPFC?

-Page 2 citation 34 does not refer to the human brain.

-Page 3 Figure 1G- this correlation is not significant and therefore conclusions should not be drawn

-Figure 2- how do they explain the lack of significant differences on the FST (ie. supplemental figure 2B)?

-SuppF3B- the authors state that no differences in the estrous cycle stage were noted, however, it looks as though almost double the number of control animals were in Metestrus. There are a number of studies showing changes in social investigation and distance travelled across the cycle, which could skew these results. Again, while it is appreciated that these factors are even considered in the current work, more probing as to the effects of cycle stage on behaviour may help to clarify effects (or non-effects) in females.

-Page 7 lines 241-6 are confusing and don't seem to match the graphs?

-Are the mouse and human cell lines from males, females or mixed?

-The authors may want to add a statement to the discussion about the EE employed. Arguably the EE employed may be considered basic housing and that the standard environment is actually an impoverished environment.

Reviewer #3

(Remarks to the Author)

I co-reviewed this manuscript with one of the reviewers who provided the listed reports. This is part of the Nature Communications initiative to facilitate training in peer review and to provide appropriate recognition for Early Career

Researchers who co-review manuscripts.

Version 1:

Reviewer comments:

Reviewer #1

(Remarks to the Author)

The authors have largely satisfied my previous concerns. I congratulate on the authors for contribution of this excellent work to NC.

Reviewer #2

(Remarks to the Author)

The revised manuscript, "Environmental enrichment and physical exercise prevent stress-induced social avoidance and blood-brain barrier alterations via Fgf2" by S.E.J. Paton, J.L. Solano, A. Cadoret, A. Collington, L. Bandeira Binder, B. Daigle, L.M. Bevilacqua, E. Richer, F. Coulombe-Rozon, L. Dion-Albert, K.A. Dudek, Signature Consortium, M. Lebel & C. Ménard examines the sex-specific protective effects of Fgf2 in attenuating stress-induced loss of the endothelial tight junction protein, Claudin-5, decreasing anxiety- and depressive-like behaviours and protection against an inflammatory challenge. In this revision, the authors have done an excellent job of addressing each of the reviewer comments and have added critical experiments that show causal evidence for the involvement of FGF2 in mediating the effects of stress resilience by directly testing knockdown of astrocytic FGF2 in the nucleus accumbens (NA). In addition, they have added appropriate controls as requested, they consider other FGF ligands, show FGF ligand expression levels and develop their rationale to focus on FGF2. They have also confirmed their FGF2 ELISA results using an appropriate high-sensitivity assay. As such, I believe that their revisions greatly strengthen the paper and their conclusions. Two minor points to edit: the AAV knockdown of FGF in Figure 5 is not huge in magnitude, and it looks like there is quite a bit of variability, where a few mice have levels comparable to controls, suggesting inadequate knockdown in these mice. As such, the authors should consider removing these mice from the behavioural studies. In addition, a correlation between FGF2 levels and behavioural outputs could be added. Finally, the NA is a small region and some images that show the extent of regional knockdown (not just cellular specificity) should be added.

Reviewer #3

(Remarks to the Author)

Open Access This Peer Review File is licensed under a Creative Commons Attribution 4.0 International License, which permits use, sharing, adaptation, distribution and reproduction in any medium or format, as long as you give appropriate credit to the original author(s) and the source, provide a link to the Creative Commons license, and indicate if changes were

made.

Response to Referees

We are pleased to submit our revised manuscript titled "**Environmental enrichment and physical exercise prevent stress-induced social avoidance and blood-brain barrier alterations via Fgf2**" for publication as an Article in Nature Communications. We addressed all comments as described in this point-by-point response and were pleased that the 3 reviewers considered our work "*interesting, clinically relevant, exciting and potentially a step forward in understanding precisely how environmental factors may protect against stress-induced effects*". The text and Figures were revised according to their recommendations (**changes in blue**), and we did perform additional experiments to reinforce our conclusions leading to the addition of 3 new Main and Supplementary Figures and revision of 3 more Figures. The Discussion was also revised to expand on limitations and refine data interpretation. The manuscript is now much stronger, and we are grateful for the careful reviews and thoughtful suggestions, which have greatly improved our study.

Reviewer 1 (Remarks to the Author):

This is an interesting study that provide new mechanistic insights into the FGF2-mediated protective effect of BBB by preventing Cldn5 loss and mitigating TNF α -mediated neuroinflammation and vascular inflammation. Exogenous delivery of AAV-FGF-2 improves depression-like behavior, inducing social avoidance. Finally, they correlated the circulating FGF-2 levels with MDD in humans and reported a significant difference between men and women and the degree of MDD severity. In general, these findings are interesting and clinically relevant. I have major concerns regarding the FGF-2 biology in relation to MDD.

We do thank the reviewer for considering our work interesting and clinically relevant.

1.1 *It seems to me that three growth factors, including VEGFA, PDGF, and FGF2, are somehow predetermined signaling molecules owing to their previously known involvements in BBB. This is a biased selection of growth factors. The authors should have deployed an unbiased approach to identify growth factors or cytokines that play key roles in their animal models, but not too much affected by what have been known.*

Response: Genes of interest in the PCR screen were determined by previous work from our group and others showing changes in expression following stress (Salmaso et al. *Biological Psychiatry*, 2016; Menard et al., *Nature Neuroscience*, 2017; Dion-Albert et al., *Nature Communications*, 2022; Matsuno et al., *Molecular Psychiatry*, 2022; Dion-Albert et al., *Research Square*, 2024). The growth factors chosen (Fgf2, Vegfa, Pdgf) are of interest since they are highly expressed by non-neuronal cells in the CNS and have been implicated in vascular pathologies and depression (Salmaso et al. *Biological Psychiatry*, 2016; Matsuno et al., *Molecular Psychiatry*, 2022; Dion-Albert et al., *Research Square*, 2024). A sentence with relevant references was added in the results section (**page 3, lines 112-113**) to clarify why these genes/growth factors were selected.

1.2 *The FGF family consists of more 20 members and some of the these members have distinct expression patterns in the brain. I am uncertain if the FGF-2 is the only FGF expressed in PFC and NAc. Given the key sex difference in their stress models, it is hard to believe that FGF-2 would execute the protective function in both cases. Other FGF members would likely to play a crucial role. It would have been really valuable if the authors looked into the differential expression profiling of growth factors in PFC and NAc areas as the initial step for the project.*

Response: Indeed, the FGF family consists of ~20 ligands, of which a subset is expressed in the brain. Most are involved in cortical patterning during development. *Fgf2* is notable for its high expression in non-neuronal cells in mice and humans (Zhang *et al.*, *J Neurosci*, 2014) (**Supp.Fig.1C**), angiogenic properties (Cao *et al.*, *Circulation Res*, 2004; Beenken *et al.*, *Nature Review Drug Disc*, 2009) and implication in MDD (Evans *et al.*, *PNAS*, 2004; Turner *et al.*, *Brain Res*, 2008; Birey *et al.*, *Neuron*, 2016; Salmaso *et al.*, *Biological Psychiatry*, 2016; Simard *et al.*, *PLoS One*, 2018; Deng *et al.*, *Front Pharmacol*, 2019). The text was revised to acknowledge existence of additional *Fgf* ligands and our rationale to focus on *Fgf2* (**page 3, lines 122-125**).

Furthermore, we assessed the expression of *Fgf1, 2, 4, 8, 9, 10,* and *21*, the most abundant of the FGF family, in the prefrontal cortex (PFC) and nucleus accumbens (NAc) of naïve male and female mice (**Supp.Fig.1D, Supp.Fig.5D**). *Fgf4, 8, 10* and *21* were undetectable in both brain areas while *Fgf9* and *10* were undetectable in cultured astrocytes and endothelial cells, our cell types of interest (**Supp.Fig.1D-E**) as expected from a brain RNA-seq atlas (Zhang *et al.*, *J Neurosci*, 2014). Importantly, stress exposure had no impact on the expression of *Fgf1* in the male NAc (**Supp.Fig.1F**). These new results were added in the revised manuscript (**page 3, lines 125-129; page 6, line 206-208**).

1.3 *FGF-2 is not a classical secretory protein. It lacks a signal peptide and remains as an intracellularly cytoplasmic protein. While the authors detected transcription expression of Fgf2 mRNA in their models, it does not mean export of FGF-2 to the extracellular space.*

Response: This is a valid point. *Fgf2* is secreted via an unconventional secretory pathway and due to the lack of a signal peptide, it is transported into the extracellular space by an ER/Golgi-independent mechanism (Steringer *et al.*, *eLife*, 2017). Immunofluorescence staining allowed us to quantify *Fgf2* protein level in the mouse NAc and PFC, but it is true that we cannot claim that *Fgf2* is secreted in the extracellular space. The Discussion was revised to briefly describe *Fgf2* unique secretory mechanism, acknowledge this limitation of our study, and suggest that future studies could take advantage of super-resolution microscopy which was used to map the FGF receptor network (Schroder *et al.*, *Methods*, 2021) (**page 22, line 638-644**).

1.4 *In the AVV system for the enforced expression of FGF-2, the authors should have introduced a signal peptide to ensure secretion of FGF-2 and have replaced the cysteine residues with other amino acids. Otherwise, the structure of FGF-2 would be changed by forming disulfide bonds.*

Response: This is an interesting suggestion. Our rationale was to mimic the endogenous increase of *Fgf2* gene expression in the NAc triggered by physical exercise (PE) or access to an enriched environment (EE). Adding a signal peptide to promote *Fgf2* secretion would certainly ensure it gets released but it may interfere with normal secretion rhythm (Murgo *et al.*, *Cell Commun Signal*, 2024). Moreover, it was shown that consistent *Fgf2* overexpression increases seizure susceptibility in transgenic mice (Zucchini *et al.*, *J Neurosci*, 2008) which would create a confounding factor for behavioral studies. This is now mentioned in the revised Discussion (**page 22, line 646-651**).

To confirm that the elevation of *Fgf2* is involved in stress resilience, we designed a new viral vector allowing astrocyte-specific downregulation of this growth factor expression. Cohorts of male mice were injected in the NAc and impact on social interactions was evaluated without or with access to physical exercise (**Fig.5**). We observed increased stress susceptibility and social avoidance in AAV5-gfaABC1D-shRNA-m*Fgf2*-Gfp-injected mice, even when they add access to running wheels, supporting our hypothesis that expression of this growth factor is involved in stress responses and benefits of PE. The text was revised to include these new findings (**page 12-14, line 339-388**).

1.5 *Detection of FGF-2 in the plasma/serum usually represents noisy backgrounds. In this case, subtle differences between 5-7.5 pg/ml should not be considered as real differences. Owing to its extremely high affinity to heparin, the circulating FGF-2 would not be detectable because the FGF-2 molecules will be trapped by interaction with heparin sulfate proteoglycans in the extracellular matrix and on the cell surface. In fact, injection of FGF2 into the blood stream results in a rapid turnover.*

Response: The reviewer is correct that quantification of FGF-2 in the plasma/serum is usually noisy. As recommended by Reviewer 2, we re-analyzed all our human serum samples with a more sensitive ELISA kit (Human FGF basic/FGF2/bFGF Quantikine high sensitive) with a detection range of 0.3-20 pg/mL and sensitivity of 0.07 pg/mL. **Fig.8C-G** panels were revised according to the new results which did not change our overall conclusions (**page 18, line 513-529**). We also added a sentence in the Discussion about the challenges related to FGF2 detection in plasma/serum and recommend highly sensitive immunoassays (**page 24, line 754-756**).

In summary, while approaching the stress disorders using experimental models is extremely important and these authors have tremendous expertise in this field, identification of signaling molecules and mechanistic insights into their functions remain very vague. Much more rigors are needed to justify their conclusions.

We are confident that the new experiments performed, including evaluation of other *Fgf* ligands' expression, design of a new viral vector downregulating astrocytic *Fgf2* and assessment of impact on social behaviors in the context of stress exposure and physical exercise, and confirmation of human FGF2 serum level changes with a highly sensitive kit, strengthen our conclusions. We thank the reviewer for recognizing relevance of our experimental approaches and expertise in the stress and BBB field and for highlighting limitations to our study that are now discussed.

Reviewer 2 (Remarks to the Author):

The manuscript, "Environmental enrichment and physical exercise prevent stress-induced social avoidance and blood-brain barrier alterations via Fgf2" by S.E.J. Paton, J.L. Solano, A. Cadoret, A. Collington, L. Bandeira Binder, E. Richer, F. Coulombe-Rozon, L. Dion-Albert, K.A. Dudek, Signature Consortium, M. Lebel & C. Ménard is a potentially exciting examination of the sex-specific protective effects of Fgf2 in attenuating stress-induced loss of the endothelial tight junction protein, Claudin-5, decreasing anxiety- and depressive-like behaviours and protection against an inflammatory challenge. The authors suggest that the protective effects of an enriched environment and voluntary physical exercise on psychological stress are, in part, mediated through an increase in the astrocyte-derived growth factor, Fgf2, which has been previously reported to have anxiolytic and antidepressant effects. Furthermore, the authors demonstrate that this increase in Fgf2 is related to BBB changes, which is known to be impaired in individuals with MDD. Changes in the BBB in MDD are clearly understudied and are likely important to both the etiology and treatment of MDD. In theory, this manuscript could provide a novel step forward in understanding precisely how environmental factors may protect against many stress-induced effects, particularly through changes in BBB function. However, some conclusions made in the manuscript are an overreach for the data presented. In addition, there are important technical limitations that need to be remediated.

We do thank the Reviewer for their positive comments and useful suggestions which we believe reinforce our conclusions.

Major:

2.1 *The authors correctly note that sex differences in MDD are marked and it is appreciated that they investigate these given the lack of studies in females in preclinical work. However, in this case, they chose to employ two different stress models, with males exposed to CSDS and females to SCVS, and then go on to make conclusions about sex differences in response to EE. CSDS and SCVS models are not interchangeable as they model different types of stress-induced depressive phenotypes. EE may have different effects in each of these models, regardless of sex and therefore conclusions cannot be made as to sex differences. Importantly, the authors again correctly note that CSDS is not easily transferable to females, however SCVS can be employed in males and therefore it is not clear why they do not include males in their SCVS study to make direct conclusions about sex differences in EE effectiveness.*

Response: This is a valid point raised by the Reviewer. To address it, a cohort of male mice was subjected to the 6-day subchronic variable stress (SCVS) paradigm with or without an enriched environment (n=6-7/group) then a battery of behavioral tests was performed like for females (**Supp.Fig.4A**). No difference in behaviors was observed due to stress (**Supp.Fig.4B-F**) in line with previous publications showing that exposure to 6-d SCVS is not sufficient to induce anxiety- and depression-like behaviors in males (*Hodes et al., J Neurosci, 2015*). A significant effect of the environment was noted for the social interaction (SI) ratio, independent of stress, due to lower time spent in the interaction zone when the CD-1 aggressor was absent for mice with access to an EE (**Supp.Fig.4C**). The text was revised to add these new findings (**page 5-6, line 192-197; page 21, line 612-614**). We also now mention in the Discussion that it would be relevant to perform a stress paradigm known to consistently induce anxiety- and depression-like behaviors in animals of both sexes, like 21-day CVS (*Labonte et al., Nature Medicine, 2017*), in future studies to enable direct comparisons of protective mechanisms (**page 21, line 628-631**).

2.2 *In several statements (including the title), the authors conclude that “Environmental enrichment and physical exercise prevent stress-induced social avoidance and blood-brain barrier alterations via Fgf2”, however they do not test whether blocking FGF2 blocks the effects of EE, PE and the BBB alterations. They show that FGF2 coincidentally goes up with PE and EE and they show that increasing FGF2 in the male NAc “mimics” the effects of PE and EE on social avoidance, however they do not show that FGF2 is causally involved in the effect of PE and EE.*

Response: As suggested, we tested whether blocking *Fgf2* expression alters social behaviors and impacts the beneficial effects of PE by designing and optimizing a new AAV downregulating *Fgf2* expression specifically in astrocytes (AAV5-gfaABC1D-shRNA-m*Fgf2*-GFP). For the current revised manuscript, we chose to focus on PE and males only, in line with the results of **Fig.3-4**. We do plan to explore the impact of manipulating *Fgf2* in the context of EE in a future study, in the NAc but also the PFC, which is more vulnerable to stress exposure in females (*Dion-Albert et al., Nature Communications, 2022*). As shown on the new **Fig.5 (page 12-14, line 339-388)**, after validation of viral efficiency (**Fig.5A-B**), male mice were injected bilaterally in the NAc with either the new virus or a control AAV5-gfaABC1D-shRNA-scramble-GFP viral vector then subjected to 10-day CSDS (**Fig.5C**). We observed increased stress susceptibility in mice injected with the virus downregulating *Fgf2* in astrocytes which was not due to impaired locomotion (**Fig.5D-F**). Another cohort of mice was injected with the same viruses and exposed to 10-d CSDS but this time, with access to running wheels (**Fig.5G**). In contrast to **Fig.3C** and **Fig.4I**, downregulation of astrocytic *Fgf2* in the NAc induced social avoidance as measured by decreased SI ratio but also increased number of entries in the corners when the aggressor was present (**Fig.5H-J**), supporting our hypothesis that an elevation of this growth factor may contribute to the benefits of PE. In fact, we noticed that stressed mice injected with the

control virus ran increasing distance across days after the first defeat bout (**Fig.5K**). This was not the case for AAV5-gfaABC1D-shRNA-mFgf2-GFP-injected mice (**Fig.5L**), suggesting that motivation could be affected, with the NAc playing a crucial role in reward processing. This is something we will explore in the future by performing behavioral tests related to cognition like the progressive ratio task, reward-based paradigms, etc. Finally, we validate AAV-driven reduced Fgf2 expression in both cohorts of mice without and with access to PE (**Fig.5M**).

2.3 *Why does FGF2 prevent social avoidance in CSDS but not SI? EE alone changes SI and social avoidance, so is FGF2 really mimicking EE? Importantly, the levels of FGF2 reported with their AAV are quite high and therefore it is striking that the behavioural effects are limited. Are changes in social avoidance behaviour alone meaningful? Is this potentially an off-target effect? For example, does increased FGF2 change activity levels in general? FGF2 KOs show decreased centre time in the open field, so is this simply a reduction in anxiety/increased exploration?*

Response: Previous studies exploring the role of Fgf2 in behavioral responses focused largely on anxiolytic effects of Fgf2 injection since it can act on glucocorticoid receptors to modify the hypothalamic-pituitary-adrenal axis activity (Perez et al., *J Neurosci*, 2009; Salmaso et al., *Biol Psychiatry*, 2016). Indeed, Simard et al. (2018) reported that FGF2 KO mice spend less time in the center of the open field test and that knocking down FGF2 has an impact on exploratory behaviour (open field, EPM) with no effect on affective behaviours as measured with sucrose consumption or the forced swim test. As recommended, we evaluated if manipulating Fgf2 could impact activity level at baseline and in the context of social stress exposure. Upregulation of Fgf2 expression in the NAc astrocytes did not alter locomotion/exploration during the SI test when the aggressor was absent with similar distance travelled for unstressed and stressed mice injected with either the control or AAV5-gfaABC1D-mFgf2-P2A-GFP virus. This important control was added on revised **Fig.3H**. Similarly, viral-mediated Fgf2 downregulation in the NAc astrocytes had no effect on locomotion/exploratory behaviors (**Fig.5F, J**). However, it could impact motivation and reward, as mentioned above, and this something we will investigate in the future with the viral tools developed here.

The Discussion was also revised to acknowledge limitations of our study regarding the unconventional secretion mechanism of Fgf2 (Steringer et al., *eLife*, 2017), and possible compensatory changes from other growth factors as reported before in transgenic mice (Miller et al., *Mol Cell Biol*, 2000; Yoshimura et al., *PNAS*, 2001; Beenken & Mohammadi, *Nature Review Drug Discov*, 2009) (**page 22, line 640-654**).

2.3 *Figure captions and methods for calculating scores need to be expanded. For example, the authors report that they use control data from past studies to compare with EE and PE. Were any controls included in the current work? How do they ensure that behavioural baselines in controls are consistent across studies?*

Response: Legends of **Fig.1-3** were expanded to provide additional information about social interaction scoring and categorization of stress-susceptible (SS) vs resilient (RES) subgroups (**page 5, line 147-150; page 10, line 277-280**). The total number of words allowed for each Figure Legend is limited to 350 with **Fig.1** and **3** currently at 351 words. We want to clarify that control unstressed groups of mice, labeled CTRL, are included in all stress paradigms and behavioral studies throughout the manuscript. The reviewer may refer to the cohorts of mice exposed to CSDS or 6-d SCVS with standard cages – no EE or PE – from our previously published studies (Menard et al., *Nature Neuroscience*, 2017; Dion-Albert et al., *Nature Communications*, 2022). These cohorts were produced using the exact same experimental protocols, defined in detail in Golden et al., *Nature Protocols*, 2011 for CSDS and Hodes et al., *J Neurosci*, 2015 for 6-d SCVS, which give reproducible results across labs worldwide. Furthermore, additional cohorts of mice with standard

cages are included in the current work for both EE (**Supp.Fig.4**), PE (**Fig.5**) and viral-mediated manipulations (**Fig.4-5**).

2.4 *It appears as though the authors used the DFB50 kit from R&D to measure FGF2 levels and not the high sensitivity (HS) kit. This kit is normed for clinical oncology samples with high levels of FGF2 and is sensitive from 10-640 pg/ml and is not sensitive for low levels of FGF2 typically seen in non-oncology samples. The HS kit is typically used for samples with low (or non-pathological) FGF2 levels as its sensitivity is between 0.3-20 pg/ml. Levels of FGF2 in Figure 7 are clearly below the sensitive range for the kit employed and therefore results may be confounded by a floor effect and conclusions cannot be drawn. Samples should be re-processed with the correct assay in order to have enough sensitivity to detect effects. Importantly, this may also help to resolve the inconsistencies in previous findings that the authors note.*

Response: We do thank the reviewer for this important suggestion. As recommended, all human blood serum samples were re-analyzed with the more sensitive ELISA kit (Human FGF basic/FGF2/bFGF Quantikine high sensitive) with a detection range of 0.3-20 pg/mL and sensitivity of 0.07 pg/mL. **Fig.8C-G** panels were revised according to the new results which did not change our overall conclusions (**page 18, line 513-529**). We also added a sentence in the Discussion about the challenges related to FGF2 detection in plasma/serum and recommend highly sensitive immunoassays (**page 24, line 754-756**).

2.5 *The data with respect to diploma and second language seems slightly out of left field. While social determinants of mental health are indeed critical, it is not clear what these two variables mean in this context?*

Response: We did aim to validate the importance of environmental conditions, as identified by hierarchical clustering in mice (**Fig.8A-B**), with socioeconomic factors available in our human cohort. Education and employment are modifiable lifestyle factors, not necessarily related to genetics, that we found could modify prevalence of MDD (**Fig.8F**). In line with this, socioeconomic position, as measured by education, income, and occupational prestige, is a strong determinant of MDD onset (*Hoveling et al. J Affect Disord, 2022*). This rationale was added to the revised manuscript (**page 18, line 516-519**). In **Supp.Table 3**, we also show data for the number of languages spoken, which did not relate to Fgf2 levels or MDD severity. Unfortunately, our cohort did not include information about housing situation or activity level.

2.6 *It might be interesting to correlate FGF2 levels to symptom clusters rather than these social determinants given previous work with FGF2 relating to specific phenotypes in humans.*

Response: This is an excellent suggestion. A new column (All) was added on the Spearman correlation heat map for both men and women MDD groups and matched healthy controls (**Fig.8G**).

Minor:

1. *How were the doses and timing of FGF2 and TNF- α chosen?*

Response: Doses of TNF- α and timepoints for cell collection were determined according to previous publications describing impact of this cytokine treatment on endothelial cells (*Cross et al., Nature; 1995; Eto et al., Circulation, 2005; Zhou et al., Am J Physiol Renal Physiol, 2008*). Cells were pretreated for 1h with Fgf2 (10 ng/mL as described in *Seghezzi et al., J Cell Biol, 1998*) to mimic habituation with EE/PE, before co-stimulation with TNF- α and/or Fgf2. These details are now clarified in the Results section (**page 14, line 404-407**).

2. *Figure 6B- it appears as though p-B-catenin is increased at 0 hours with TNF α and not with TNF α + FGF2? Also, what does CTRL refer to?*

Response: In now **Fig.7A-B**, CTRL refers to 0h with no FGF2 pretreatment. The Figure panels and Legend were modified to add this precision (**page 17, line 483-487**). Control lanes are darker in **Fig.7B** for the TNF- α treated vs. TNF- α + FGF2 treated gels, however, brightness is arbitrary for both figures and analysis is relative to CTRL (not between gels).

3. *The authors refer to the PFC generally but may want to specify as it looks as though they are investigating the mPFC?*

Response: This is a good point. We used PFC to be consistent with our previous publications (*Dion-Albert et al., Nature Communications, 2022; Dudek et al., Nature Neuroscience, 2025*).

4. *Page 2 citation 34 does not refer to the human brain.*

Response: This is right, Ref. 34 (*Zhang et al., J Neurosci, 2024*) refers to mouse RNA seq only. Thus, the 2016 human RNA seq reference was added (*Zhang et al., Neuron, 2016*) (**page 2, line 72-75**).

5. *Page 3 Figure 1G- this correlation is not significant and therefore conclusions should not be drawn*

Response: We do not mention a significant correlation but only a trend (now **page 5, line 168-171**).

6. *Figure 2- how do they explain the lack of significant differences on the FST (ie. supplemental figure 2B)?*

Response: Female mice were subjected to a battery of behavioral tests as described in our previous publication (*Dion-Albert et al., Nature Communications, 2022*). Forced swim test (FST) was the last test performed and 6-day SCVS is considered a mild stressor, so it is possible that the stress effect had worn off at that time. Moreover, the validity of FST is increasingly contested as a screen for depressive-like behaviors and it could be more associated with stress coping strategies (*Commons et al., ACS Chem Neurosci., 2017*).

7. *SuppF3B- the authors state that no differences in the estrous cycle stage were noted, however, it looks as though almost double the number of control animals were in Metestrus. There are a number of studies showing changes in social investigation and distance travelled across the cycle, which could skew these results. Again, while it is appreciated that these factors are even considered in the current work, more probing as to the effects of cycle stage on behaviour may help to clarify effects (or non-effects) in females.*

Response: We do agree with the reviewer that it will be important to investigate further how hormonal regulation, notably the estrus cycle and testosterone level, can affect behavioral responses with sex hormones known to be potent modulators of BBB integrity. This is something we have been advocating for in the brain barriers field as highlighted in these publications: *Dion-Albert et al., Front Neuroendocrinol, 2022; Collignon et al., Fluids Barriers CNS, 2024*. The Discussion was revised accordingly (**page 21, line 626-628**).

8. Page 7 lines 241-6 are confusing and don't seem to match the graphs?

Response: The text was revised to better describe the findings. The sentence “*This implies that while EE alters more broadly the transcriptional stress response at the neurovasculature, PE more precisely targets Cldn5 expression*” was removed while “*At protein level, CSDS did not affect Cldn5 levels in the NAc of our PE cohort, whereas all stressed PE mice exhibited an increase in Fgf2 staining when compared to controls*”, was revised (now page 8, line 266-268).

9. Are the mouse and human cell lines from males, females or mixed?

Response: The HBEC-5i cell line is from a male donor (source: https://www.cellosaurus.org/CVCL_4D10) (page 36, line 1283-1285). This information is unfortunately not available for the bEnd.3 cell line.

10. The authors may want to add a statement to the discussion about the EE employed. Arguably the EE employed may be considered basic housing and that the standard environment is actually an impoverished environment.

Response: The Discussion was revised as recommended (page 21, line 593-600).

Reviewer 3 (Remarks to the Author):

Response to Referees

We are pleased to submit our revised manuscript titled "**Environmental enrichment and physical exercise prevent stress-induced social avoidance and blood-brain barrier alterations via Fgf2**" for publication as an Article in Nature Communications. We addressed all remaining reviewers' comments, as described in this point-by-point response.

Reviewer 1 (Remarks to the Author):

The authors have largely satisfied my previous concerns. I congratulate on the authors for contribution of this excellent work to NC.

Reviewer 2 (Remarks to the Author):

The revised manuscript, "Environmental enrichment and physical exercise prevent stress-induced social avoidance and blood-brain barrier alterations via Fgf2" by S.E.J. Paton, J.L. Solano, A. Cadoret, A. Collington, L. Bandeira Binder, B. Daigle, L.M. Bevilacqua, E. Richer, F. Coulombe-Rozon, L. Dion-Albert, K.A. Dudek, Signature Consortium, M. Lebel & C. Ménard examines the sex-specific protective effects of Fgf2 in attenuating stress-induced loss of the endothelial tight junction protein, Claudin-5, decreasing anxiety- and depressive-like behaviours and protection against an inflammatory challenge. In this revision, the authors have done an excellent job of addressing each of the reviewer comments and have added critical experiments that show causal evidence for the involvement of FGF2 in mediating the effects of stress resilience by directly testing knockdown of astrocytic FGF2 in the nucleus accumbens (NA). In addition, they have added appropriate controls as requested, they consider other FGF ligands, show FGF ligand expression levels and develop their rationale to focus on FGF2. They have also confirmed their FGF2 ELISA results using an appropriate high-sensitivity assay. As such, I believe that their revisions greatly strengthen the paper and their conclusions.

Two minor points to edit: the AAV knockdown of FGF in Figure 5 is not huge in magnitude, and it looks like there is quite a bit of variability, where a few mice have levels comparable to controls, suggesting inadequate knockdown in these mice. As such, the authors should consider removing these mice from the behavioural studies. In addition, a correlation between FGF2 levels and behavioural outputs could be added.

The reviewer is possibly referring to **Fig.5B** in which *Fgf2* knockdown was confirmed in astrocytes in a cohort of mice that did not go through behavioral testing. The text was revised to clarify (**page 6, line 262-265**). For behavioral studies, viral-mediated downregulation of astrocytic *Fgf2* was confirmed by qPCR (**Fig.5M**). As expected, most mice are stress-susceptible in the AAV5-gfaABC1D-shRNA-m*Fgf2* group (17/24) without or with access to running wheels (**Fig.5D and 5H**) and accordingly, *Fgf2* is down-regulated by 20-50% in 22/24 animals. With so few resilient animals and a bias toward stress susceptibility driven by the viral manipulation, we do not believe that correlative studies are appropriate. However, we should mention that we do observe a positive correlation between *Fgf2* levels and social interaction ratio for resilient mice (*p=0.0222) and a trend for unstressed controls (*p=0.0547) when the viruses are combined.

Finally, the NA is a small region and some images that show the extent of regional knockdown (not just cellular specificity) should be added.

As recommended, **Fig.5B** was revised to add lower magnification images of NAc viral infection.

Reviewer 3 (Remarks to the Author):
